# Immediate hydroxylation of arenes to phenols via V-containing all-silica ZSM-22 zeolite triggered non-radical mechanism

Yu Zhou[1], Zhipan Ma[1], Junjie Tang[1], Ning Yan [2], Yonghua Du [3], Shibo Xi[3], Kai Wang[1], Wei Zhang[1], Haimeng Wen[1] & Jun Wang[1]

Hydroxylation of arenes via activation of aromatic $C_{sp2}$–H bond has attracted great attention for decades but remains a huge challenge. Herein, we achieve the ring hydroxylation of various arenes with stoichiometric hydrogen peroxide ($H_2O_2$) into the corresponding phenols on a robust heterogeneous catalyst series of V–Si–ZSM-22 (TON type vanadium silicalite zeolites) that is straightforward synthesized from an unusual ionic liquid involved dry-gel-conversion route. For benzene hydroxylation, the phenol yield is 30.8% (selectivity >99%). Ring hydroxylation of mono-/di-alkylbenzenes and halogenated aromatic hydrocarbons cause the yields up to 26.2% and selectivities above 90%. The reaction is completed within 30 s, the fastest occasion so far, resulting in ultra-high turnover frequencies (TOFs). Systematic characterization including $^{51}$V NMR and X-ray absorption fine structure (XAFS) analyses suggest that such high activity associates with the unique non-radical hydroxylation mechanism arising from the in situ created diperoxo V(IV) state.

---

[1] State Key Laboratory of Materials-Oriented Chemical Engineering, College of Chemical Engineering, Nanjing Tech University (former Nanjing University of Technology), Nanjing 210009, P.R. China. [2] Department of Chemical and Biomolecular Engineering, National University of Singapore, 4 Engineering Drive 4, Singapore 117585, Singapore. [3] Institute of Chemical and Engineering Sciences, 1 Pesek Road, Jurong Island, Singapore 627833, Singapore. These authors contributed equally: Yu Zhou, Zhipan Ma. Correspondence and requests for materials should be addressed to N.Y. (email: ning.yan@nus.edu.sg) or to J.W. (email: junwang@njtech.edu.cn)

Aromatic compounds with hydroxyl groups in the benzene ring are a group of important organic intermediates widely applied as chemical precursors of dyes, polymers, plastics, pharmaceuticals, and agrochemicals[1–12]. Even for the simplest group member phenol, it is industrially produced through the multi-step cumene process with low yield (~5%), which is energy consuming and environmentally unfavorable[4,6]. Direct one-step hydroxylation of the benzene ring with economically and environmentally benign oxidants such as hydrogen peroxide ($H_2O_2$) is one of the most promising alternatives for the production of phenol and its derivatives[8,10,13–20]. However, this process faces two major challenges: (1) low catalysis efficiency due to the inherent inertness of the aromatic ring $C_{sp2}$–H bonds and the undesirable but inevitable over-oxidation of the product phenols;[20–22] (2) preferential side-chain oxidation of the $C_{sp3}$–H bonds in substituted arenes like toluene rather than aromatic ring oxidation of the $C_{sp2}$–H bonds with higher bond dissociation energies[23,24].

For decades, direct hydroxylation of arenes with $H_2O_2$ has been broadly explored in both homogeneous and heterogeneous catalysis over numerous catalysts such as zeolites[17,25], poly-oxometallates[23,26], carbon[19], and metal complexes[27,28]. Recently, very high turnover numbers (TON) and turnover frequencies (TOF), as well as high selectivity for aromatic ring oxidation of substituted arenes, were achieved by homogeneous catalysts[20,23,24,29]. Heterogeneous catalysis is preferred due to the facile catalyst separation and reuse[8,10,13–17,25,26], but the aforementioned challenges are more difficult to overcome. One prime cause is that non-uniformity of a solid surface always leads to diverse catalytic active sites with different chemical composition, geometric configuration, or coordination environment. As a result, heterogeneous catalysts usually demonstrated inferior activities with lower TOF, particularly with lower selectivity for the oxidation of substituted benzene derivatives[8,17,26]. Also, this will raise more complexities in bringing insight into the heterogeneous reaction mechanism.

Vanadium (V) is among the most widespread transition metals to catalyze the hydroxylation of arenes with $H_2O_2$[8,16,23,26,30]. Its abundant coordination mode enables versatile activities but significantly complicates the catalytic active species, especially when employed as a heterogeneous catalyst[31–34]. In the V-catalyzed hydroxylation of aromatic ring with $H_2O_2$, the initial and key step is the reduction of the high-valent V(V) species by $H_2O_2$ into the low-valent V(IV) site, followed with the creation of intermediate hydroxyl radicals on V(IV)[26,35]. Because $H_2O_2$ is intrinsically an oxidation agent rather than a reductant, that initial step is slow. Further, the strong oxidation property of the resulted hydroxyl radicals also accounts for the over-oxidation of phenols due to their non-regio-selective feature[14,18,36]. Thus all the early attempts aiming to significantly enhance the reaction rate and selectivity were hindered by the conventional radical mechanism. Constructing uniformly dispersed, unconventional non-radical mechanistic active V sites on a solid catalyst is an effective way to tackle above problems, but scarcely reported before.

Herein, we fabricate an efficient heterogeneous catalytic system for the direct hydroxylation of arenes (benzene, alkylbenzenes, and various substituted derivations) with $H_2O_2$ based on V-containing all-silica zeolite (V–Si–ZSM-22). ZSM-22 is a TON topological aluminosilicate zeolite material with a ten-member-ring one-dimensional pore system[37]. The V–Si–ZSM-22 series in this work is task-specifically designed and straightforward synthesized from an unusual dry-gel-conversion (DGC) route. Systematic catalysis assessments for the hydroxylation of benzene/toluene showed that the reaction proceeded efficiently under stoichiometric condition (arene/$H_2O_2$ = 1/1) with high yield/selectivity, stable recyclability and broad substrate compatibility.

Full catalyst characterization including [51]V NMR and XAFS spectra indicate the uniform dispersion of V species on the zeolite framework. More interestingly, analysis of the intermediate catalyst phases reveals a rapid generation of an active V peroxo species for selective aromatic ring hydroxylation, evidencing a non-radical mechanism. As a result, all the hydroxylations could be completed almost immediately (less than 30 s), giving an ultra-high TOF. The result is in clear contrast to previous hydroxylation systems with much longer reaction time of several to tens of hours even mostly using $H_2O_2$ to arene substrates ratio far away from stoichiometry[13,14,20,26,29,32]. As we are aware, this is the best catalytic system, in terms of TOF and space time yield, for the direct hydroxylation of benzene and substituted benzene derivatives based on a heterogeneous catalyst.

## Results

**Catalyst synthesis and characterization.** V–Si–ZSM-22 was synthesized in an ionic liquid templated DGC process, in which the dry gel was prepared through the co-hydrolysis/condensation of tetraethylorthosilicate (TEOS) and ammonium metavanadate ($NH_4VO_3$) under mild acidic conditions (pH = ~1.0, Fig. 1)[38,39]. Varying the molar ratio of V to Si in the gel produced a series of V–Si–ZSM-22 samples, termed VSZ-$n$ ($n = 100 \times$ [V/Si molar ratio in the gel]). Their V content was close to that in the gel (Supplementary Table 1), thanks to the DGC process that inhibits the V leakage. Therefore, V content is facilely controlled through modulating the gel composition. XRD patterns (Supplementary Fig. 1) demonstrated the well crystal structure of TON topology for the samples with the $n$ up to 25[37]. Further increasing the V content in the gel caused amorphous structure (Supplementary Fig. 2). No signal for any impurity phases was detected, excluding the formation of vanadium oxide ($V_2O_5$) crystals. The unit cell parameters of VSZ-$n$ are similar to that of the V-free counterpart Si–ZSM-22 (Supplementary Table 2), suggesting non-existence of isomorphously substituted V ions in the zeolite framework.

SEM images showed that Si–ZSM-22 was composed of intertwined thin sheets in micrometer level, while the primary particles of VSZ-$n$ exhibited the similar shape with a mat surface at low V contents ($n = 1$ and 3) and gradually evolved to be small pomegranate grain-like blocks with apparent fusion at high V contents (Fig. 2a and Supplementary Fig. 3). Elemental mapping images of the typical sample VSZ-5 presented relatively homo-dispersion of the V species (Supplementary Fig. 4). TEM images (Fig. 2b–d) revealed the ordered intracrystal microporous channels, index of well crystal structure that is further reflected by [29]Si NMR spectrum (Supplementary Fig. 5)[37,39]. No $V_2O_5$ crystals were observable, implying the formation of amorphous V species. FT-IR spectra confirmed their TON type framework, within which IL template remained evidently in the as-synthesized samples (Supplementary Figs. 6, 7)[37]. The type I feature of the nitrogen sorption isotherms for VSZ-$n$ (Supplementary Fig. 8) demonstrated their typical microporous structure[37,40]. With the increase of V content, their surface area and pore volume (Supplementary Table 1) slightly decreased in the case of $n = 1$–5 and rapidly declined at high V contents ($n = 10$–25), reflecting the partial pore blockage by extra-framework V species. Thermogravimetric (TG) curves for VSZ-$n$ (Supplementary Fig. 9) indicated high thermal stability and resistance to moisture due to the strong hydrophobicity derived from their all-silica skeleton[37,41].

UV-vis spectra of as-synthesized VSZ-$n$ demonstrated two main peaks centered at 260 and 340 nm (Supplementary Fig. 10) for the $\pi$(t2)$\rightarrow$d(e) and $\pi$(t1)$\rightarrow$d(e) oxygen-tetrahedral V(V) charge transfer (CT) transitions, involving bridging (V–O–Si or V–O–V) and terminal (V=O) oxygen[38,42]. The weak bands at 500–800 nm

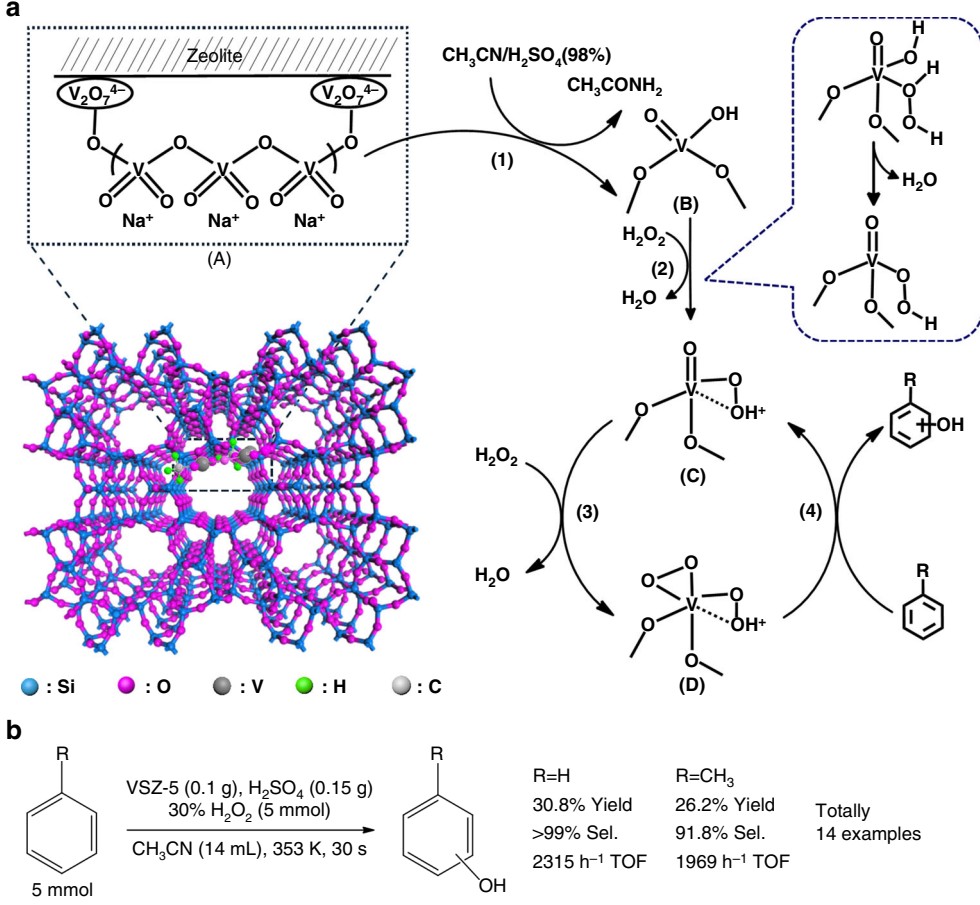

**Fig. 1** Structure and activity of V–Si–ZSM-22 in the hydroxylation of arenes. **a** Structure and proposed reaction mechanism. **b** Reactivity of hydroxylation of arenes with stoichiometric $H_2O_2$ catalysed by VSZ-5

come from the V(IV) (3d1) species[39], in accordance with sharp ESR signals at 3000–4000 G (Supplementary Fig. 11)[39,43,44]. After calcination, this d–d band disappeared (Fig. 2e); also, the ESR signal became almost silent (Supplementary Fig. 12) while a sharp peak at 517.1 eV ($V2p_{3/2}$) occurred in the V2p XPS spectrum (Supplementary Fig. 13). These observations suggest that rare V (IV) species exist in the calcined samples[26,39,45]. Raman spectra of VSZ-$n$ (Fig. 2f) exhibited strong bands for V=O (921, 957 $cm^{-1}$) and V–O–V (512, 640 $cm^{-1}$) of the metavanadate chains, suggesting the formation of oligomeric metavanadic groups $(VO_3)_n^{n-}$ with tetrahedral coordination[46]. The weak band at 885 $cm^{-1}$ for $V_2O_7^{4-}$ species[34,39] was observable for VSZ(1–5). The proportion of the $V_2O_7^{4-}$ species decreased at high V contents ($n = 10$–25), reflected by the even weaker intensity of the 885 $cm^{-1}$ band. Similar variation happened on the band at 785 $cm^{-1}$ for the Si–O–V. These phenomena reveal the formation of $V_2O_7^{4-}$ species bonded to the zeolite framework, particularly in the case of low V contents. Accordingly, these $V_2O_7^{4-}$ species act as the bridge covalently linking $(VO_3)_n^{n-}$ and zeolite silica skeleton (Fig. 1 and Supplementary Fig. 14). $^{51}V$ MAS NMR spectra of typical samples VSZ (1–10) (Fig. 2g) verified the structure of V species and the variation with their contents. Each sample presented a sharp symmetric peak at −576 ppm attributable to the tetrahedral metavanadate V(V) species $(VO_3)_n^{n-}$ and a 559 ppm shoulder to $V_2O_7^{4-}$ species[47–49]. Declined intensity of 559 ppm signal at high V contents again reflects the decrease of $V_2O_7^{4-}$ species. XAFS analysis of VSZ-5 further revealed that these V species have similar coordination environment to those in $NH_4VO_3$ (Fig. 2h–j and Supplementary

Figs. 15, 16), reflecting the formation of the relatively uniform diamagnetic V(V) species in tetrahedral metavanadic state[49,50]. The fitting results of extended X-Ray absorption fine structure (EXAFS) spectra (Fig. 2j and Supplementary Table 3) demonstrated two short bonds (V=O, 1.66 Å) and two long bonds (V–O, 1.82 Å). The counter cations are $Na^+$ as demonstrated by the chemical composition in Supplementary Table 1. All the spectra confirm the structure in Fig. 1a. $H_2$-TPR curves (Supplementary Fig. 17) presented a high-temperature peak for $H_2$ desorption, suggesting the strong host-guest interaction between these inside-channel V species and zeolite skeleton[39]. By contrast, the low-temperature desorption peak emerging on the samples at high V contents ($n \geq 10$) is assignable to the reduction of V species outside the channels with weak host-guest interaction.

**Catalytic activity in hydroxylation of arenes**. Catalytic assessments of VSZ-$n$ samples were parallel conducted in the $H_2O_2$-mediated hydroxylation of benzene to phenol and toluene to cresols under the stoichiometric condition (arene/$H_2O_2$ = 1/1) (Figs. 1b, 3 and Supplementary Tables 4, 5). For hydroxylation of benzene, the reaction did not happen without a catalyst and only trace amount of product was detected over V-free Si-ZSM-22 (Supplementary Table 4, entry 1–3). By contrast, the reaction was efficiently catalyzed by VSZ-$n$, giving >99% phenol selectivity (Fig. 3a and Supplementary Table 4, entry 4–11). The phenol yield firstly rose with the increase of V content and then decreased at the high V content. The highest phenol yield reached 30.8% over VSZ-5 (the sample simultaneously with relatively high

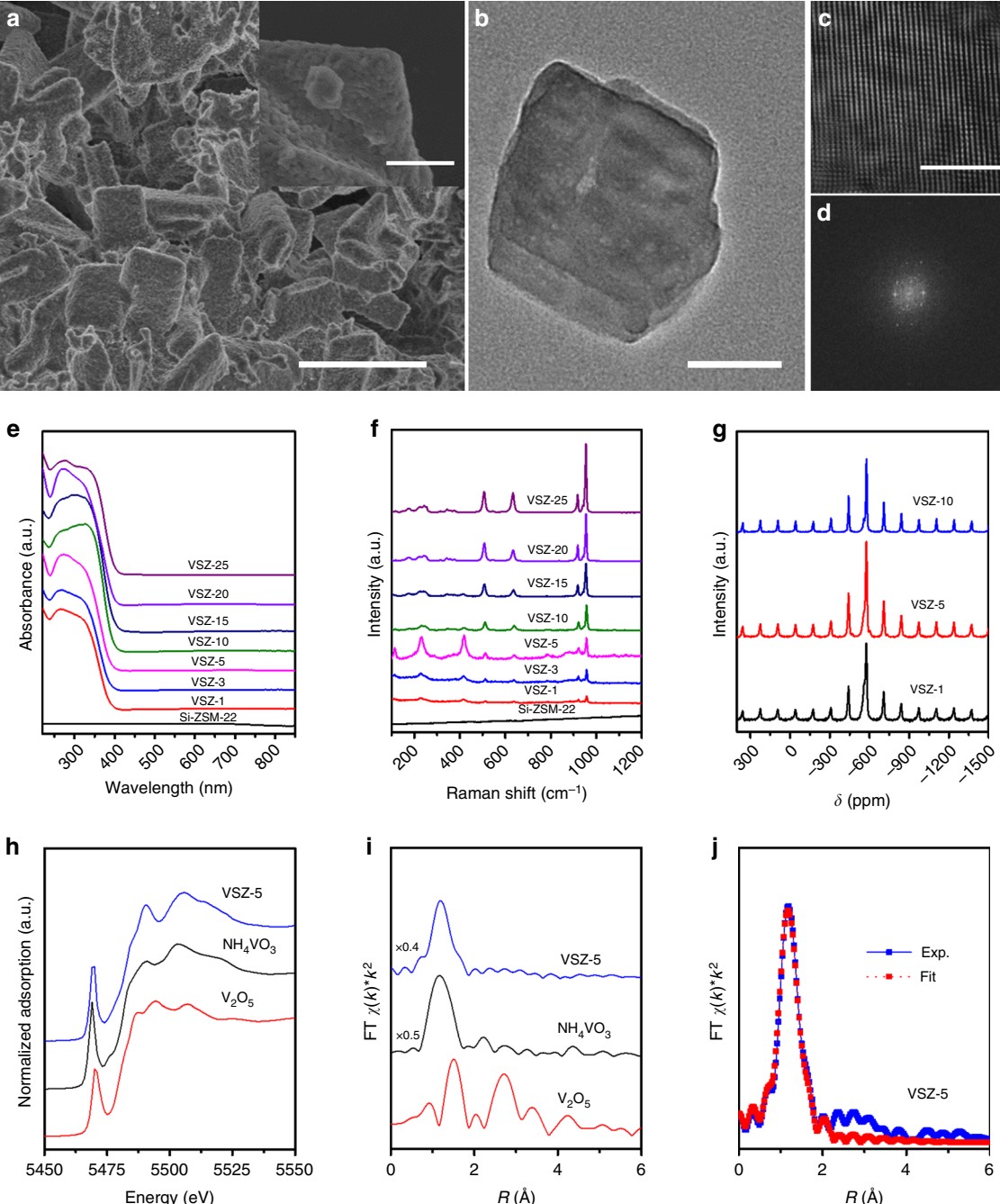

**Fig. 2** Structure characterization. **a** SEM and **b–d** TEM images of VSZ-5; **e** UV-vis, **f** Raman, and **g** $^{51}$V MAS NMR spectra of VSZ-$n$ samples; **h** X-ray absorption near edge structure (XANES) of V K-edge for VSZ-5 with the reference materials $NH_4VO_3$ and $V_2O_5$; **i** the $k^2$-weighted Fourier transform spectra derived from EXAFS for VSZ-5 with the reference materials $NH_4VO_3$ and $V_2O_5$; **j** FT-EXAFS curves between the experimental data and the fit. Scale bars, 10 μm and insert diagram of 2 μm (**a**), 50 nm (**b**), and 5 nm (**c**)

V content and large surface area) within a short reaction time (Fig. 3a, b). Selective ring hydroxylation of toluene was also achieved over VSZ-$n$ to produce cresols with almost equal $o$- and $p$-product (Fig. 3d and Supplementary Table 5). VSZ-5 gave the highest cresols yield of 26.2% and selectivity of 91.8% (Fig. 3d, e), rendering an efficient heterogeneous catalyst for the highly chemo-selective oxidation of toluene to cresols. A hot-filtration test was performed by filtrating the solid catalyst after the addition of $H_2SO_4$. The filtrate solution is inactive in promoting benzene to phenol, verifying that the hydroxylation process is intrinsically heterogeneous and excludes the contribution of the potentially leached V species. After reaction, VSZ-5 can be facilely recovered and reused. In the six-run recycling tests, stable activity was found in the hydroxylation of benzene and toluene (Fig. 3c, f), revealing excellent catalyst reusability.

Various conditions influencing VSZ-5 catalyzed hydroxylation of benzene/toluene were investigated (Fig. 3b, e and Supplementary Figs. 18–28). Both reactions completed immediately after the addition of $H_2O_2$ (we added the preset amount of aqueous $H_2O_2$ into reaction media within 1 min), while longer reaction time caused negligible variation of the activity (Fig. 3b, e). The reaction time was thus fixed at 30 s, considering that even shorter reaction time was not practical. The reaction time scale in seconds was greatly shorter than previous catalytic systems (normally in hours

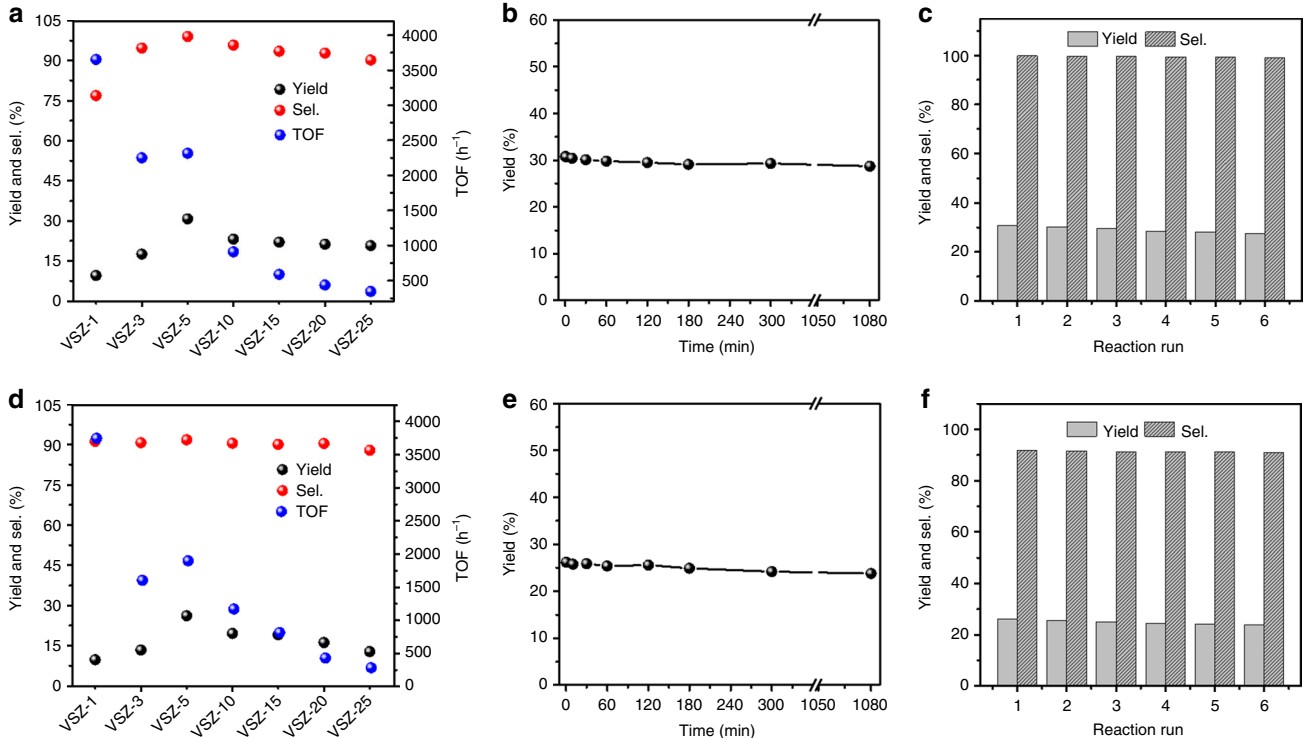

**Fig. 3** Catalysis performance of VSZ-n. Yield, selectivity and TOF of VSZ-n in the hydroxylation of **a** benzene and **d** toluene; Yield as a function of reaction time in the VSZ-5 catalyzed hydroxylation of **b** benzene and **e** toluene; Catalytic reusability of VSZ-5 in the hydroxylation of **c** benzene and **f** toluene. Reaction conditions: substrate (5 mmol), $H_2O_2$ (30%, 5 mmol), $H_2SO_4$ (0.15 g), $CH_3CN$ (14 mL for benzene; 12 mL for toluene), 80 °C, 30 s (for **a**, **c**, **d**, and **f**)

or longer, Supplementary Tables 6, 7). Correlatively, VSZ-5 offered an ultra-high TOF up to 2315 and 1969 $h^{-1}$ for benzene and toluene hydroxylations, respectively, dramatically exceeding previous heterogeneous catalysts and even superior to the most effective homogeneous one (Supplementary Tables 6, 7). If a low amount of catalyst was used, a maximum TOF of 5707 $h^{-1}$ could be obtained in VSZ-1 catalyzed benzene hydroxylation (Supplementary Table 4, entry 5). In most previous efforts, employing non-stoichiometric amount of $H_2O_2$ with respect to the substrates was a necessity to gain the desired aromatic ring hydroxylation (Supplementary Tables 6, 7). Significantly in our case, VSZ-5 catalyzed reaction happened with the substrate/oxidant ratio of 1:1, rendering the high atom-efficiency of both substrate and oxidant.

The activation energy barrier in the VSZ-5 catalyzed hydroxylation of benzene/toluene is calculated according the Arrhenius Equation: $k = A*\exp(-Ea/RT)$ ($k$: reaction rate; $A$: pre-exponential factor; $Ea$: activation energy; $R$: gas constant; $T$: reaction temperature). Because the reaction was finished within a short time (30 s), we assume that the reaction belongs to a linear kinetic curve ($d[phenol]/dt = k$), just like the very initial stage of a conventional kinetic curve for the benzene hydroxylation with the reaction time in hours. Therefore, the reaction rate at different temperature is calculated from $[phenol]/t$, in which $[phenol]$ is the phenol concentration after reaction while $t$ is fixed as 30 s. According to this assumption and based on the activity data in Supplementary Figs. 18, 24, the linear plot of $\ln k$ vs. $1/T$ was performed and the slope is $-Ea/R$ (Supplementary Fig. 29). The calculated Ea is 24 and 26 kJ $mol^{-1}$ for the VSZ-5 catalyzed hydroxylation of benzene and toluene, respectively. The relatively low activation energies are in accordance with the rapid reaction rate in these hydroxylation processes. Kinetic curves were analyzed for the VSZ-5 catalyzed benzene hydroxylation. The fitting results suggest that the reaction rate is proportional to

concentration of catalyst VSZ-5, benzene and $H_2O_2$ (Supplementary Figs. 30–32).

Several control catalysts were tested in the oxidation of benzene and toluene, including neat $V_2O_5$, Si–ZSM-22 supported $V_2O_5$ prepared by wet-impregnation ($V_2O_5$@ZSM-22), V-containing aluminosilicate zeolite (V–AlSi–ZSM-22), plus the homogeneous catalyst $NH_4VO_3$ (their structural information was shown in Supplementary Table 1 and Supplementary Figs. 33–39). When $NH_4VO_3$ was dissolved in acetonitrile ($CH_3CN$), various V species were formed in the solution[49]. These control solid catalysts also have various V species on the surface. All of them exhibited inferior activity in the hydroxylation of both benzene (yields: 10.6–15.4%; selectivities: 76.5–87.1%, Supplementary Table 4, entry 12–15) and toluene (yields: 8.3–11.9%; selectivities: 70.8–83.9%, Supplementary Table 5, entry 10–13). Such comparisons confirm the superior performance of VSZ-5, implying that the uniform tetrahedral V(V) species of the polymeric metavanadic groups on the framework of all-silica zeolite ZSM-22 is highly active for the aromatic ring hydroxylation.

The scope of VSZ-5 was extended to the hydroxylation of other arenes with stoichiometric $H_2O_2$ (Table 1). Various mono-/di-alkylbenzenes and halogenated aromatic hydrocarbons were chemo-selectively oxidized to the corresponding phenols. Same as the situation of toluene hydroxylation, the aromatic $sp^2$ carbon in all those alkylbenzenes was preferentially oxidized with the yields for phenols of 20.6–24.1%. Despite the fact that the alkylbenzenes contain more reactive aliphatic C–H bonds in the side-chain, the selectivies for aromatic ring hydroxylation were all above 90%. Such high chemo-selectivity for alkylbenzenes other than benzene is hardly achieved over the most effective homogeneous catalysts even under the specific condition of a low molar ratio of $H_2O_2$/substrate. For example, the osmium(VI) nitrido catalyst was highly selective for the hydroxylation of alkylbenzenes but far less selective for benzene substrate[24]. It is

**Table 1 Direct hydroxylation of various arene substrates with H$_2$O$_2$ catalyzed by VSZ-5.†**

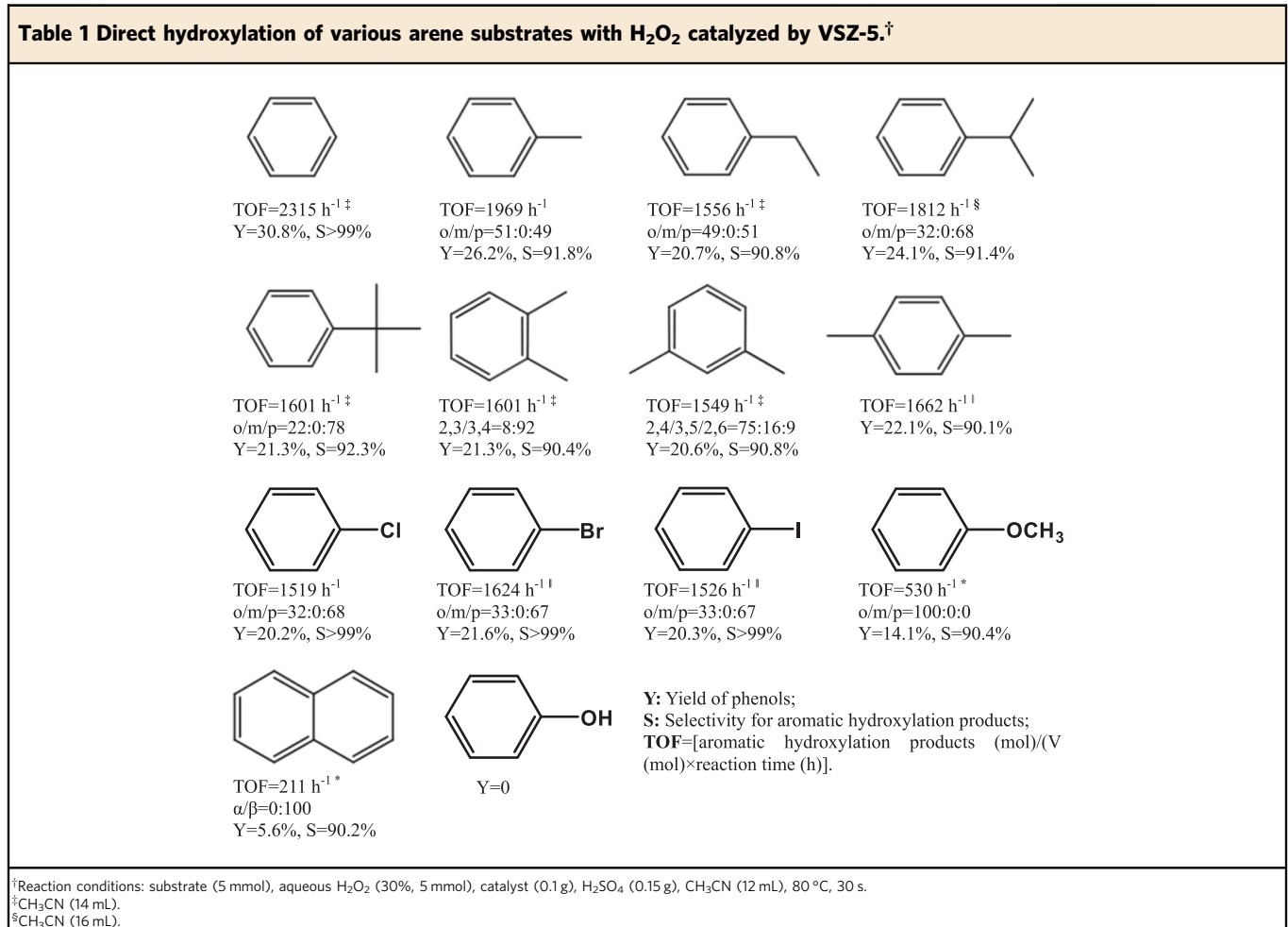

TOF=2315 h$^{-1}$‡
Y=30.8%, S>99%

TOF=1969 h$^{-1}$
o/m/p=51:0:49
Y=26.2%, S=91.8%

TOF=1556 h$^{-1}$‡
o/m/p=49:0:51
Y=20.7%, S=90.8%

TOF=1812 h$^{-1}$§
o/m/p=32:0:68
Y=24.1%, S=91.4%

TOF=1601 h$^{-1}$‡
o/m/p=22:0:78
Y=21.3%, S=92.3%

TOF=1601 h$^{-1}$‡
2,3/3,4=8:92
Y=21.3%, S=90.4%

TOF=1549 h$^{-1}$‡
2,4/3,5/2,6=75:16:9
Y=20.6%, S=90.8%

TOF=1662 h$^{-1}$‖
Y=22.1%, S=90.1%

TOF=1519 h$^{-1}$
o/m/p=32:0:68
Y=20.2%, S>99%

TOF=1624 h$^{-1}$‖
o/m/p=33:0:67
Y=21.6%, S>99%

TOF=1526 h$^{-1}$‖
o/m/p=33:0:67
Y=20.3%, S>99%

TOF=530 h$^{-1}$*
o/m/p=100:0:0
Y=14.1%, S=90.4%

TOF=211 h$^{-1}$*
α/β=0:100
Y=5.6%, S=90.2%

Y=0

**Y:** Yield of phenols;
**S:** Selectivity for aromatic hydroxylation products;
**TOF**=[aromatic hydroxylation products (mol)/(V (mol)×reaction time (h)].

†Reaction conditions: substrate (5 mmol), aqueous H$_2$O$_2$ (30%, 5 mmol), catalyst (0.1 g), H$_2$SO$_4$ (0.15 g), CH$_3$CN (12 mL), 80 °C, 30 s.
‡CH$_3$CN (14 mL).
§CH$_3$CN (16 mL).
‖CH$_3$CN (14 mL), 82 °C.
*Aqueous H$_2$O$_2$ (30%, 2.5 mmol); Yield of phenols: [phenols (mol)/initial H$_2$O$_2$ (mol)] × 100.

more interesting for the inert aryl halides with strong electron-withdrawing groups, the high chemo-selectivity for aromatic ring hydroxylation was similarly obtained, with phenols yields of 20.2–21.6% and selectivities larger than 99%. Anisole with strong electron-donating side-chain and bulky naphthalene were also chemo- and region-selectively oxidized into *o*-hydroxyanisole and 2-hydroxynaphthalene, respectively. All the reactions were fulfilled within 30 s, affording much higher TOF values (>1000 h$^{-1}$) than previous systems (Supplementary Tables 6, 7)[20,23,24,28,29]. These results suggest wide substrate compatibility for the selective activation of aromatic C$_{sp2}$–H bond. No *m*-product and coupling product was detectable for the mono-substituted benzenes, reflecting a non-radical hydroxylation character. *p*-Product was predominately formed for the large-sized substrates of benzene derivatives; the selectivity for the *p*-product was 78% in the hydroxylation of tert-butylbenzene. Considering that there exist two *o*-positions and one *p*-position in a benzene ring, the probability was ~88% to attack the *p*-position in the tert-butylbenzene hydroxylation, revealing an obvious shape-selective catalysis process. Such regio-selectivity comes from the size effect of the narrow ten-member-ring microporous channels of TON framework, in which the diffusion of *o*-product with larger dynamitic size than *p*-product is limited. The shape-selectivity is further reflected by the hydroxylation of naphthalene, in which only the small-sized product, 2-hydroxynaphthalene, could be formed due to the limitation of mass transfer for other bulky ones.

All the above reactions were conducted with CH$_3$CN as the solvent and sulfuric acid (H$_2$SO$_4$) as the additive. The influence of acid additives and solvents was studied in the VSZ-5 catalyzed toluene hydroxylation (Supplementary Table 5, entry 14–26). H$_2$SO$_4$ itself was inert (yield: 0%, Supplementary Table 5, entry 2), while only 1.8% yield with 56.9% selectivity (entry 14) was observed in the absence of H$_2$SO$_4$. Other acids including ascorbic acid (Vc), acetic acid (CH$_3$COOH), trifluoromethanesulfonic acid (CF$_3$SO$_3$H), perchloric acid (HClO$_4$) and hydrochloric acid (HCl) were explored by using CH$_3$CN as the solvent (entry 15–21). Besides, other common solvents including water (H$_2$O), methanol (CH$_3$OH), ethanol (CH$_3$CH$_2$OH), dimethylformamide (DMF), benzonitrile (C$_6$H$_5$CN), acetone (CH$_3$COCH$_3$) and dimethyl sulfoxide (DMSO) were tested in the presence of H$_2$SO$_4$ (entry 22–26). However, none of them afforded comparable yield of phenol, indicating that the conjunction of H$_2$SO$_4$ and CH$_3$CN plays a crucial role in the reaction.

The color of the VSZ-5 solid in the CH$_3$CN solvent changed from white to dark green after the addition of H$_2$SO$_4$. By contrast, the color change was not observed by using other acids or solvents, and rare hydroxylation happened after further addition of H$_2$O$_2$. This phenomenon indicates that the dark green intermediate, denoted as VSZ-5(m), is essential for the reaction. We collected VSZ-5(m) by filtration and made systematic characterization (Supplementary Figs. 5 and 40–57). XRD, SEM, TEM, $^{29}$Si NMR, and nitrogen sorption analyses revealed that VSZ-5(m) retained the initial crystal structure and porosity of its mother sample VSZ-5. Sharp ESR peaks at 3000–4000 G (Supplementary Fig. 45) and V2p$_{3/2}$ XPS signal shifting to 516.2 eV (Supplementary Fig. 47) suggested the formation of V(IV)

species in VSZ-5(m)[39,43], agreeing with the color variation. The suggestion is confirmed by the silent signal of the $H_2$-TPR curve (Supplementary Fig. 48). Further, [51]V NMR spectrum of VSZ-5 (m) presented only a weak peak at 559 ppm for $V_2O_7^{4-}$ species (Supplementary Fig. 44), implying that the tetrahedral V(V) species in the form of $(VO_3)_n^{n-}$ were reduced to V(IV) species after adding $H_2SO_4$. Raman spectrum (Supplementary Fig. 43) displayed no signal at 957 nm attributable to the V=O group[46], reflecting the disappearance of initial O=V=O structure in VSZ-5(m). XAFS analysis (Supplementary Figs. 54–56 and Supplementary Table 3) revealed that VSZ-5(m) mainly contained V (IV) species with tetrahedral coordination[50], in which there existed one short bond (V=O, 1.60 Å) and three long bonds (V–O, 2.02 Å). These results prove the structure variation from V (V) to V(IV) species after the addition of $H_2SO_4$ (step (1), Fig. 1a). GC and GC-MSD analysis of the filtrate after removal of VSZ-5(m) indicated the formation of about 0.019 mmol acet-amide ($CH_3CONH_2$) per gram VSZ-5 (0.4 mmol V). Such observation reveals that in the first step of Fig. 1a, the V(V) species of $(VO_3)_n^{n-}$ in VSZ-5 were transformed into V(IV), whilst $CH_3CONH_2$ was created from $CH_3CN$[51]. No $CH_3CONH_2$ was detectable in the absence of VSZ-5 or by using other acids/ solvents, implying that this transformation only happened in the simultaneous presence of VSZ-5/$CH_3CN$/$H_2SO_4$. In a separate run, the hydroxylation of toluene with $H_2O_2$ directly catalyzed by the isolated VSZ-5(m) in the absence of $H_2SO_4$ produced the corresponding cresols in the considerable yield of 17.1% with the selectivity of 89.2% (Supplementary Table 5, entry 27), strongly supporting that the in situ formed V(IV) species indeed contribute to the catalytic activity. This is further reflected by the positive correlation between the number of the V(IV) species and the reactivity (Supplementary Fig. 46).

Systematic characterization of the recovered catalysts was carried out (Supplementary Figs. 5 and 40–59). The recovered catalyst after the first run, denoted as VSZ-5(r1), well preserved the crystal structure and high dispersion of V species (Supplementary Figs. 5 and 40–58). UV-vis (Supplementary Fig. 42), Raman (Supplementary Fig. 43), [51]V NMR (Supplementary Fig. 44), and XAFS spectra (Supplementary Figs. 54, 55, 58 and Supplementary Table 3) indicate that these V species of VSZ-5(r1) vary into the umbrella model (Supplementary Fig. 60)[52,53]. To probe variation of the V species during the first run, the recovered VSZ-5 without calcination, termed VSZ-5(r1)-as, was collected and characterized. At the end of a reaction, the green color of the catalyst solid in the reaction solution suggests that its V species stayed in the V(IV) state, which is further corroborated by the strong ESR peak of VSZ-5(r1)-as (Supplementary Fig. 61)[39,43,44]. Raman spectrum of VSZ-5(r1)-as displayed an apparent adsorption peak around 915 nm, indicative of the V-oxo species (Supplementary Fig. 62)[52,53]. The result suggests that, after reaction, the V species have changed into the oxo umbrella structure rather than returned to the state of VSZ-5(m) or the fresh catalyst. VSZ-5(r1)-as alone was still active for the reaction. The activity was 30 or 63% relative to that of the fresh catalyst in the absence or presence of $H_2SO_4$ (Supplementary Fig. 63), respectively. This phenomenon implies that the umbrella V(IV) species play a key role in the hydroxylation process. After calcination, the green color of the V(IV)-bearing VSZ-5(r1)-as sample changed to the white color of the V(V)-containing VSZ-5 (r1) (Supplementary Figs. 45,47,48). When VSZ-5(r1) was engaged into the second run, the catalyst color changed again to dark green (i.e., recurrence the intermediate VSZ-5(m)) after the addition of $H_2SO_4$; and after the further addition of $H_2O_2$, same hydroxylation performance was obtained as the fresh one. The recovered catalyst after sixth run, VSZ-5(r6), demonstrated almost the same structure as VSZ-5(r1) (Supplementary Figs. 40–55 and

59 and Supplementary Table 3), accounting for the stable reusability. Moreover, no apparent variation of the V content was observed over the reused catalyst and the leached V species in the solution was undetectable, meaning that the V species is robustly grafted on the zeolite skeleton. The small amount of $V_2O_7^{4-}$ species in VSZ-5, though not the active sites, serve as the bridge to link the $(VO_3)_n^{n-}$ active sites and silica framework (Fig. 1a and Supplementary Fig. 14), and therefore endow high stability.

## Discussion

Kinetic isotope effect (KIE, $k_H/k_D$) for the VSZ-5 catalyzed hydroxylation of benzene was determined by competitive hydroxylation of $C_6H_6$ and $C_6D_6$ with $H_2O_2$. The intramolecular KIE value was 1.13 for this hydroxylation. Such low KIE value excludes the involvement of the C-H bond cleavage in the rate-determining step[20,24,28,35]. This is consistent with the fact that there was no hydroxylation product due to the NIH shift (also known as 1,2-hydride shift; the name NIH shift arises from the US National Institutes of Health (NIH) that first reported this transformation) accompanying with the formation of carbocations via C–H bond cleavage. Additionally, the KIE value suggests that the hydroxyl radical and H-atom abstraction are not involved, as the reported KIE values for these two mechanisms were 1.7 and 4.9, respectively[20,24,28]. V peroxy radicals are also unlikely, as metal–O–O· active sites usually caused high NIH shift[35], the preferential side-chain oxidation of toluene into benzaldehyde[54] or low activity in the hydroxylation of haloge-nated aromatics[29]. Radical scavenger tests were performed and no deactivation happened in the presence of tert-butyl alcohol (TBA, scavenger for hydroxyl radicals), butylated hydroxytoluene (BHT, scavenger for superoxide radicals), benzoquinone (BQ, scavenger for superoxide radicals), 5,5-Dimethyl-1-pyrroline N-oxide (DMPO, radical trapping reagent for various radicals), or bro-motrichloromethane ($BrCCl_3$, scavenger for carbon-centered radicals) (Supplementary Fig. 64). All these observations exclude the possibility that the hydroxylation would have involved a radical route, suggesting that the VSZ-5 catalyzed hydroxylation reaction may involve the formation of metal-active oxygen species.

Previously, electrophilic metal-oxygen species, such as metal=O, metal–O–O· and metal bis(μ-oxo), have been proposed in several non-radical hydroxylation reactions[20,23,24,28,29]. In these hydroxylation processes, high activity was observed by using the mono-substituted benzenes with electron-donating groups such as anisole and phenol. By contrast, apparent high activity in the hydroxylation of benzene and suppression of the activity in the hydroxylation of anisole and phenol was observed using the VSZ-5 catalyst. Besides, for aryl halides substrates with electron-withdrawing groups, the high activity and chemo-selectivity was similarly obtained in the current study, which was rarely achieved in those previous electrophilic reactions. These phenomena suggest that our V-based hydroxylation pro-cess is different from those previous systems involving electro-philic attack[20,23,24,28,29].

Structure characterization of the fresh VSZ-5 and the inter-mediate VSZ-5(m) indicated that the tetrahedral metavanadate V (V) species $(VO_3)_n^{n-}$ in the fresh catalyst changed to V(IV) species (**B**) after the addition of $H_2SO_4$. Previous studies of the behavior of V species in the presence of $H_2O_2$ suggested that the species (**B**) is readily converted into (**C**), quickly via the two transition states marked in the prompting frame in Fig. 1a[35,55–59]. The observed umbrella model V species in the recovered catalysts VSZ-5(r1) and VSZ-5(r6) as well as the characterization of VSZ-5 (r1)-as support the formation of this V(IV)-based peroxo

compound during the reaction. The structure characterization and activity assess of VSZ-5(r1) and VSZ-5(r1)-as implies that the arene was oxidized by the active metal-oxygen sites derived from species (**C**) in the presence of $H_2O_2$. In a non-radical hydroxylation reaction, further interaction of $H_2O_2$ with (**C**) favors to create V(IV) diperoxo species (**D**) through the formation of a $H_2O_2$-$V^{IV}$ complex, followed by H-transfer to the oxo-ligand and elimination of $H_2O$ (Supplementary Fig. 65)[44,58,59]. The formed diperoxo group in species (**D**) is a highly active oxygen transfer agent to initiate the ring hydroxylation of arenes[35,44,55–60].

Based on above, we propose a probable reaction mechanism in Fig. 1a. V(V) species (**A**) of the fresh VSZ-n in $CH_3CN$ was reduced to V(IV) species (**B**) in the intermediate VSZ-5(m) upon addition of $H_2SO_4$. After introducing $H_2O_2$, (**B**) was readily converted into V(IV) peroxo complex (**C**) and then (**D**). The interaction of species (**D**) with arene forms a transition complex, the cleavage of which affords phenol and regenerates species (**C**). According to this mechanism, the kinetic equation is given as follows: $d[phenol]/dt = k'[Cat.][S][H_2O_2]$ ([phenol], [Cat.], [S], and [$H_2O_2$] are the concentration of phenol, catalyst, substrate and $H_2O_2$, respectively; $t$ is the reaction time; $k'$ is the catalytic rate constant). This equation is in accordance with the experimental kinetic equation (Supplementary Figs. 30–32), thus further supporting the above proposed mechanism.

The reaction of V-peroxo compound with arene to produce phenol may undergo either homolysis or heterolysis of V–O bond[35,55,58,60]. According to the previous studies related to the oxygen transfer from V-peroxo species to arenes[35,44,55–60], we tentatively propose that a heterolytic cleavage of V–O in species (**D**) may occur in the oxidation of benzene ring (Supplementary Fig. 66). Thus, formed $V^V$–O–$O^-$ species are different from the previous metal=O and metal–O–O· species[20,23,24,28,29]. The $V^V$ cation of $V^V$–O–$O^-$ is a Lewis's acid site and able to interact with the negative π-system of the benzene ring via polarization. Such polarization interaction will promote the approaching and adsorption of arene, and the successive nucleophilic attack of $O^-$ species on the C and H to form a transition complex, the cleavage of which produces phenol and regenerates species (**C**). Such special behavior is attributable to the formation of V(IV) peroxo compounds that lead to an unusual oxidation process. Methoxy and hydroxyl group are electron-donating groups as they have the strong resonance effect (though with the coexistence of weak inductive effect). Thus, the hydroxylation of anisole and phenol is not favored through the above proposed mechanism, in line with the experimental results. Noticeably, this attacking mode particularly favors high chemo-selectivity by avoiding the over-oxidation of the hydroxylated products phenols.

According to this potential non-radical mechanism, no reduction of the V(V) to V(IV) by $H_2O_2$ is needed to initiate the reaction, accounting for the dramatically enhanced overall reaction rate. The switch between (**C**) and (**D**) is a rapid step to enable a fast reaction rate[35,55–60]. The reaction was completed within 30 s, while further elongating the time (even up to 1080 min) caused no apparent decline of the phenol yield (Fig. 3b, e), implying successful avoiding of over-oxidation. The phenomenon is distinct from previous over-oxidation induced volcanic type kinetic curves[14,36]. Briefly, the non-radical mechanism ensures quick and selective activation of aromatic $C_{sp2}$–H bonds and inhibits the over-oxidation of phenols. On the other hand, the highly dispersed $(VO_3)_n^{n-}$ species in the fresh catalyst and then the quick creation of V(IV) intermediates in the reaction system provide relatively uniform active sites for the reaction. By contrast, even for the homogeneous $NH_4VO_3$ catalyst, various V species are formed when it is dissolved in the solvent[49], explaining its lower yield and selectivity than VSZ-5. Similarly, owing to the existence of different V species on the surface, $V_2O_5$ and $V_2O_5$@Si–ZSM-22 exhibited

low activity. The low activity of Al-containing counterpart V–AlSi–ZSM-22 suggests that the all-silica framework in VSZ-5 benefits the reaction by reducing the surface non-uniformity. Besides, the all-silica framework enables surface super-hydrophobicity that favors the affinity of organic substrates and enhances the reaction efficiency.

While the proposed mechanism as discussed above is based on a number of experiments and spectroscopic analysis, it only represents one possible reaction pathway that requires further investigation in the future. For instance, the inertness of hydroxylation of phenol is still unclear, as the electron-donating property of hydroxyl group is normally recognized as close to that of methoxy group.

In summary, immediate hydroxylation of arenes to phenols with stoichiometric $H_2O_2$ was achieved employing vanadium silicalite zeolites with TON topology, which were straightforwardly synthesized from an ionic liquid assisted DGC route. The salient features of the present catalytic system include high yield/selectivity, stable reusability and broad substrate compatibility. Further, the short reaction time enabled ultra-high TOF values and space time yield, benefiting the potential industry application. The in situ formed relatively uniform peroxo V(IV) species allowing an unusual non-radical reaction mechanism play a key role toward such high performance. This work provides a promising approach toward robust heterogeneous catalysis with extremely rapid reaction rate and high activity for the selective $H_2O_2$-based hydroxylation of $C_{sp2}$–H bonds.

## Methods

**Materials and synthesis**. All chemicals were analytical grade and used as received. TON type vanadium silicalites V–Si–ZSM-22 were synthesized in a DGC route. In a typical synthesis, aqueous sulfuric acid solution (25.26 g, pH = 1.0), tetraethylorthosilicate (TEOS, 7.40 g) and calculated amount of ammonium metavanadate ($NH_4VO_3$) was mixed in a 100-mL glass beaker and stirred at room temperature for 24 h. Ionic liquid (IL) 1-butyl-3-methylimidazolium bromide ([BMIm]Br) and NaOH solution (12.5 mol $L^{-1}$, 1.70 g) was successively added. The gel with the molar composition of $n$% $NH_4VO_3$: 1 $SiO_2$: 0.35 [BMIm]Br: 0.2 $Na_2O$: 40 $H_2O$ ($n$ = 1, 3, 5, 10, 15, 20, and 25) was aged at room temperature under vigorous stirring for 24 h, and then dried at 373 K for 4 h. The obtained dry gel was crystallized at 443 K for 2 days in a 50 mL Teflon-lined steel autoclave with a raised Teflon holder inside to place the gel (0.5 g) and water (0.5 g) in the upper and bottom, respectively. After crystallization, the solid was collected, washed with ethanol and water, and dried at 373 K for 12 h to give the as-synthesized samples. Template was removed through the calcination at 873 K for 5 h in air, giving the as-calcined products, named as VSZ-n, in which $n$ = 100 × [V/Si molar ratio in the gel].

V-free counterpart Si-ZSM-22 was synthesized with the same procedure in the absence of $NH_4VO_3$. Vanadium oxide ($V_2O_5$) supported on Si-ZSM-22 ($V_2O_5$@Si-ZSM-22), V- and Al-containing ZSM-22 (V–Al–ZSM-22) were prepared, with the details in the Supplementary Methods.

**Characterization**. X-ray diffraction (XRD) analysis was performed on a SmartLab diffractmeter (Rigaku) equipped with a 9 kW rotating anode Cu source (45 kV, 200 mA, 5–50°, 0.2°/s). Scanning electron microscope (SEM) and corresponding elemental mapping images were taken from field-emission scanning electron microscope (Hitachi S-4800). Transmission electron microscopy (TEM) and corresponding elemental mapping images were carried out with field-emission transmission electron microscope JEM-2100F (200 kV). Nitrogen sorption experiments were measured at 77 K on a BEL SORP-MAX analyzer. Before measurements, samples were outgassed at 573 K to a vacuum of $10^{-3}$ Torr. Fourier transform infrared spectroscopy (FT-IR) spectra ranging 4000–400 $cm^{-1}$ were recorded on an Agilent Cary 660 instrument (KBr disks). Magic angle spinning nuclear magnetic resonance (MAS-NMR) spectra were collected at a Larmor frequency of 105.181 MHz and a magnetic field strength of 9.4 T using a 4.0 mm MAS probe with spinning at 12 kHz ($^{29}$Si-NMR) and 14 kHz ($^{51}$V-NMR) on a Bruker Avance III spectrometer. The metal content was determined by using ADVANT'XP X-ray fluorescence (XRF) spectrometer (ZSX Primus II). Diffuse reflectance UV-vis spectra were recorded on a SHIMADZU UV-2600 spectrometer by using barium sulfate ($BaSO_4$) as the internal standard. X-ray photoelectron spectroscopy (XPS) analysis was conducted on a PHI 5000 Versa Probe X-ray photoelectron spectrometer equipped with Al-Kα radiation (1486.6 eV). Electron spin resonance (ESR) spectra were recorded on a Bruker EMX-10/12 spectrometer at X-band at room temperature. Thermogravimetric (TG) curves were collected on a STA 409 instrument in air (10 °C $min^{-1}$). Raman spectra were recorded on a Jobin Yvon

(Laboratory RAM HR1800) confocal micro-Raman spectrometer backscattered geometry through a 10× (NA = 0.25) microscope objective. Hydrogen temperature-programmed reduction (H$_2$-TPR) curves were determined on Catalyst Analyzer BELCAT-B under a gaseous mixture of 10% H$_2$/Ar (30 mL min$^{-1}$) from 40 to 800 °C (10 °C min$^{-1}$). H$_2$ consumption was monitored by a thermal conductivity detector (TCD). The sample was pre-treated under argon gas flow at 200 °C for 2 h. V Kedge XAFS of various V catalysts and reference materials of V$_2$O$_5$, VO$_2$, V$_2$O$_3$, NH$_4$VO$_3$, and vanadium acetylacetonate (VAcAt) were recorded at XAFCA beamline of Singapore Synchrotron Light Source (SSLS) with electron energy of 0.7 GeV[61]. The X-ray energy was calibrated at the inflection point of the absorption edge of metallic vanadium. Analyses of X-ray absorption near-edge spectra (XANES) and extended X-ray absorption fine structure (EXAFS) spectra were conducted by using Athena and Artemis included in the Demeter package. Each theoretical scattering path was generated with FEFF 6.0 L for the curve-fitting analysis on EXAFS spectra. Fourier transformation of the $k^2$-weighted EXAFS oscillation was carried out in the range of 2.2–11 Å$^{-1}$ [62].

**Catalytic activity evaluation.** Direct hydroxylation of arenes with hydrogen peroxide (H$_2$O$_2$, 30% aqueous solution) was conducted in a 25 mL quartz glass tube reactor equipped with a magnetic stirrer. In a typical run of hydroxylation of toluene, 0.1 g catalyst was mixed with 5 mmol toluene and 12 mL solvent acetonitrile (CH$_3$CN). Concentrated sulfuric acid (0.15 g) and 5 mmol H$_2$O$_2$ was successively dropwise added under vigorous stirring. Reaction proceeded at 353 K. The identification of products was analyzed by GC/MSD (Agilent Technologies 7890B-5977A GC/MSD) equipped with a capillary column (HP-5MS 30 m × 0.32 mm × 0.25 μm) and a Triple-Axis detector. Quantitative analyses were carried out by gas chromatography (GC, Agilent 7890B) equipped with a flame ionization detector and a capillary column (HP-5, 30 m × 0.25 mm × 0.25 μm). Typical GC and GC/MSD spectra are illustrated in Supplementary Fig. 67. The major products for the selective oxidation of toluene were o- and p-cresols. The detected by-product was benzaldehyde generated from the oxidation of the side chain, while other possible by-product like benzylalcohol was in trace amount. Hydroxylation of other arenes was performed similarly by using the target substrate. After reaction, the solid catalyst was recovered by filtration, washed, and calcined at 550 °C for 5 h. The recovered catalyst was reused in the subsequent run without adding fresh one. A hot-filtration test was performed to confirm the heterogeneous nature of the hydroxylation reaction. Because the reaction is finished within a short time, it is impossible to hot-filtrate the catalyst during the reaction. The operation was slightly different from the traditional test. Details are as follows. The catalyst VSZ-5 (0.1 g) was mixed with benzene (5 mmol), CH$_3$CN (14 mL) and H$_2$SO$_4$ (0.15 g). After the slurry mixture was stirred at 80 °C for 30 min, the catalyst was removed by filtration. The reaction of the filtrate solution alone was carried out with the addition of 5 mmol H$_2$O$_2$ (30% aqueous solution) at 80 °C. The products were measured by GC.

**Data availability.** The data that support the findings of this study are either providing in the Article and its supplementary information or are available from the authors upon reasonable request.

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

## Acknowledgements

This work was supported by the National Natural Science Foundation of China (Nos. 21476109, U1662107, 21136005, and 21303084), Specialized Research Fund for the Doctoral Program of Higher Education (No. 20133221120002) and the Project of Priority Academic Program Development of Jiangsu Higher Education Institutions (PAPD). J.W. and Y.Z. thank the support of Jiangsu National Synergetic Innovation Center for Advanced Materials. N.Y. thanks the Young Investigator Award from National University of Singapore (R-279-000-464-133) for the financial support.

## Author contributions

J.W. conceived and supervised the project and revised the manuscript. N.Y. co-supervised the project and co-designed the experiments. Y.Z. designed the experiments, analyzed most of the data, and wrote the manuscript. Z.M. carried out most syntheses, characterization, and catalysis evaluation. J.T. carried out partial syntheses, characterization, and catalysis evaluation. Y.D. and S.X. carried out the XAFS measurements. K.W., W.Z., and H.W. took part in materials syntheses and/or characterization. All the authors discussed the results and commented on the manuscript.

## Additional information

**Competing interests:** The authors declare no competing interests.

