## [Peer Review File · Nature Communications]

Reviewers' comments:

Reviewer #1 (Remarks to the Author):

The article “Immediate hydroxylation of arenes to phenols: V-containing all-silica ZSM-22 Zeolite triggered non-radical mechanism” by Wang et.al. Although the manuscript reports high activity and the catalyst is well characterized, however, there are several major drawbacks and I cannot recommend this manuscript for the publication in Nature Communications.

1. Authors use Conc. Sulphuric acid for the activation of reactive vanadium species; however, they completely ignored the activation of benzene ring due to Conc. Sulphuric acid, which may lead them a different kind of mechanism. The proposed reaction mechanism is too speculative. I would like to suggest authors should rethink about their proposed mechanism or can provide some spectroscopic evidences to support their claim.
2. Authors used H₂O₂ with very less amount (arene /H₂O₂ ration) in their reaction and comparatively conversion levels are very high in a very short time, So efficiency of H₂O₂ for this catalyst is quite high , which needs to be crosschecked!
3. Above all the colour change of the catalyst from white to green after addition of Conc. Sulphuric acid to the reaction mixture showing the leaching tendency of Vanadium species during the reaction. This indicates that the catalyst is not truly heterogeneous. The colour of the catalyst should be white for truly heterogeneous in nature.

Reviewer #2 (Remarks to the Author):

This work described a nice catalyst for hydroxylation of arenes. The extremely high selectivity at an elevated conversion is certainly significant for production of phenols in a green way. As the authors emphasize the non-radical mechanism, I cannot accept this work to be published before several obsessions are well documented.

- 1.The reduced VO₃ (IV) seems to be considered as the real active sites. Then, a positive correlation between the number of the active sites and the reactivity. Perhaps by ESR?
2. Since the reaction is finished very quickly (30 S), then why not increase the amount of substrates to get a more precise TOF?
- 3.The reaction time contributed by diffusion effect is notable considering the microporous

structure of ZSM. Then the actual activity should well be higher than the offered value. With such a high reaction rate, it is hard to understand the non-radical mechanism. A apparent energy barrier is necessary as a reference. If the calculated energy barrier is lower than 30 KJ/mol. Then it means the diffusion is controlled. With such a high activity to C-H activation, it is hard to understand the extremely low activity to phenol over-oxidation in that the electron-withdrawing strength of OH group is between OCH₃ and H, as mentioned by the authors about the substituent effect in Table 1.

4.As for Fig 3b and e, it seems unnecessary to prolong the reaction time since no additional oxidant (H₂O₂) is left and the reaction stop naturally.

5.As for the radical scavenger tests, can they diffuse into the micro-pores?

6.For the step from (B) to (C) in Fig 1, why H₂O can served as a stable ligand without being abstracted by one H by the atop O binding with V, namely the usual hydrogen transfer effect?

7.For the step from (C) to (D) in Fig 1, the authors should give a more detailed description about the process since I cannot map this route without radical pathway.

Reviewer #3 (Remarks to the Author):

This is an interesting report. The system reported by the authors seem to be highly effective and selective for the conversion of benzene to phenol, an important industrial reaction. While I am not an expert on the characterization of the catalyst my analysis of the data would suggest that this system is unusually active compared to other reports. While the author makes such a comparison with respect to TOF, selectivity, yields, etc., I was perhaps most impressed by the volumetric productivity, Space Time Yield (STY). Assuming 5 mmol of product in 14 ml over 30 s I calculate a STY of ~ 10E-5 mol/cc.s. This is 10 times higher than the typical STY of many commercial processes based on the so-called Weiss window of 10-6 mol/cc.s. Initially, I was concerned that H₂SO₄ additive was responsible for the catalysis but the authors address this. The only issue I have with this report is that the suggested basis for the selectivity does not seem reasonable. It is remarkable that phenol and anisol are less reactive than benzene. This is certainly good evidence against a radical mechanism. However, other metal based systems, including those based on V, do not show such remarkable selectivity. The authors suggest that this is due to the O on the active V(IV)O species being “negative” and that OMe and OH are donor groups that inhibit attack on phenol or anisol. This does not seem reasonable. Nucleophilic attacks on aromatics leading to oxidation typically require nitro groups to activate the arene. It is unlikely that benzene would enter in such a nucleophilic reaction. In any case, while the O in V(IV)-O is formally 2-, it is not considered a nucleophile and likely reacts as an O-electrophilic. Such species have been proposed for Os, Cu and V based on homogeneous systems. Additionally, while OMe and OH are pi-donors (they facilitate aromatic electrophilic substitution through resonance not through inductive effects) they are sigma-acceptors (more electron-withdrawing than H) and should activate anisol or phenol to nucleophilic reaction. This unusual

lack of reaction with phenol bothers me but I cannot see an issue based on the data. I was struck by what seems to be a considerable amount of “hype” in the report. I recommend this be removed as it is distracting at best and over-selling at worst. I will leave that up to the editor. I recommend publication.

Responses to referees

(Manuscript ID: NCOMMS-17-33106)

Reviewer #1:

1. Authors use Conc. Sulphuric acid for the activation of reactive vanadium species; however, they completely ignored the activation of benzene ring due to Conc. Sulphuric acid, which may lead them a different kind of mechanism. The proposed reaction mechanism is too speculative. I would like to suggest authors should rethink about their proposed mechanism or can provide some spectroscopic evidences to support their claim.

Answer: Many thanks for the helpful comments from the reviewer. The concentrated H₂SO₄ was admixed by acetonitrile and water (from the aqueous H₂O₂) forming a diluted acid reaction system during catalysis. Indeed, the acid concentration was only ~0.11 mol L⁻¹ in the typical benzene hydroxylation reaction (5 mmol benzene, 5 mmol 30% H₂O₂, 0.15 g H₂SO₄, 14 mL CH₃CN). As mentioned in the seventh paragraph of the section “Catalytic activity: Hydroxylation of arenes”, the H₂SO₄-CH₃CN solution caused the transformation of the V(V) species of (VO₃)_nⁿ⁻ in VSZ-5 into the V(IV) species in VSZ-5(m) (the first step of Fig. 1a). Isolated VSZ-5(m), in the absence of H₂SO₄, afforded a yield of 17.1% in the hydroxylation of toluene, suggesting that the *in-situ* formed V(IV) species in VSZ-5(m) was able to catalyze the hydroxylation of arene without the help of H₂SO₄. To gain a deeper insight into reaction mechanism, we determined the kinetic isotope effect (KIE, k_H/k_D) for the VSZ-5 catalyzed hydroxylation of benzene by competitive hydroxylation of C₆H₆ and C₆D₆ with H₂O₂. KIE value was 1.13, indicating that the C-H bond cleavage is not included in the rate-determining step. Therefore, in this work, the major function of H₂SO₄ (together with CH₃CN solvent) is to create the V(IV) active species for initiating the catalysis cycle, while the direct activation of benzene ring due to the presence of H₂SO₄ did not play a significant role.

The reaction mechanism was proposed based on the catalytic performance of VSZ-5 in the hydroxylation reactions (relative reactivity and regio-selectivity for benzene and its substituted derivatives), collection and characterization of the intermediate VSZ-5(m) as well as the recovered catalysts (VSZ-5(r1) and VSZ-5(r6)), radical scavenger tests, and analysis of related literatures. We are sorry that we did not specially emphasize this information that confused the reviewer. Besides, we newly collected/analyzed the following data: (1) analysis of KIE value (as mentioned in the paragraph above); (2) two other scavengers including butylated hydroxytoluene (BHT, scavenger for superoxide radicals) and 5,5-dimethyl-1-pyrroline N-oxide (DMPO, radical trapping

reagent for various radicals) were applied, aiming to rule out the possible radical-related reaction pathways (as shown in Fig. R1); (3) kinetic analysis based on existing experimental results (influence of catalyst amount and H₂O₂ amount in the VSZ-5 catalyzed benzene hydroxylation, Supplementary Figs 21 and 22) and additional activity assessment (influence of substrate amount in VSZ-5 catalyzed benzene hydroxylation, Fig. R2) and (4) ESR analysis to semi-quantify the formation of V(IV) species, based on which a positive correlation between the number of the V(IV) species and the reactivity was established (details are seen in the Answer to Question 1 of Reviewer #2). Combining these, we reconsider the mechanism and modify the discussions to give a clearer image of how the system works. Details are presented below.

Figure R1 | Radical scavenger tests. VSZ-5 catalyzed direct hydroxylation of toluene in the presence of tert-butyl alcohol (TBA, scavenger for hydroxyl radicals), butylated hydroxytoluene (BHT, scavenger for superoxide radicals), benzoquinone (BQ, scavenger for superoxide radicals), 5,5-Dimethyl-1-pyrroline N-oxide (DMPO, radical trapping reagent for various radicals), or bromotrichloromethane (BrCCl₃, scavenger for carbon-centered radicals). Reaction conditions: toluene (5 mmol), aqueous H₂O₂ (30%, 5 mmol), catalyst (0.1 g), H₂SO₄ (0.15 g), CH₃CN (12 mL), 80 °C, 30 s.

The intramolecular KIE value was determined to be 1.13 for VSZ-5 catalyzed benzene hydroxylation. Such low KIE value excludes the involvement of the C-H bond cleavage in the rate-determining step (*Angew. Chem. Int. Ed.* 56, 12260–12263 (2017); *J. Am. Chem. Soc.* 137, 5867–5870 (2015); *Chem. Commun.* 50, 11454–11457 (2014); *J. Am. Chem. Soc.* 105, 3101–3110 (1983)). This is consistent with the fact that there was no hydroxylation product due to the NIH shift (also known as 1,2-hydride shift; the name NIH shift arises from the US National Institutes of Health (NIH) that first reported this transformation) accompanying with the formation of carbocations *via* C-H bond cleavage. Additionally, this KIE value suggests that the hydroxyl radical and H-atom abstraction are not involved, as the reported KIE values for these two mechanisms were 1.7 and 4.9 respectively (*Angew. Chem. Int. Ed.* 56, 12260–12263 (2017); *J. Am. Chem. Soc.* 137, 5867–5870 (2015); *Chem. Commun.* 50, 11454–11457 (2014)). V peroxy radicals

are also unlikely, as metal-O-O· active sites usually caused high NIH shift (*J. Am. Chem. Soc.* 105, 3101–3110 (1983)), the preferential side-chain oxidation of toluene into benzaldehyde (*J. Mol. Catal. A: Chem.* 253 1–7 (2006)) or low activity in the hydroxylation of halogenated aromatics (*Angew. Chem. Int. Ed.* 56, 7779–7782 (2017)). Radical scavenger tests were performed and no deactivation happened in the presence of tert-butyl alcohol (TBA, scavenger for hydroxyl radicals), butylated hydroxytoluene (BHT, scavenger for superoxide radicals), benzoquinone (BQ, scavenger for superoxide radicals), 5,5-Dimethyl-1-pyrroline N-oxide (DMPO, radical trapping reagent for various radicals), or bromotrichloromethane (BrCCl₃, scavenger for carbon-centered radicals) (Fig. R1). All these observations exclude the possibility that the hydroxylation would have involved a radical route, suggesting that the VSZ-5 catalyzed hydroxylation reaction may undergo the formation of metal-active oxygen species.

Previously, electrophilic, non-radical aromatic substitution mechanism was proposed for Os, Cu and V based homogeneous systems (also mentioned by Reviewer #3) (*Angew. Chem. Int. Ed.* 56, 12260–12263 (2017); *Angew. Chem. Int. Ed.* 56, 7779–7782 (2017); *J. Am. Chem. Soc.* 137, 5867–5870 (2015); *Chem. Commun.* 50, 11454–11457 (2014); *Angew. Chem., Int. Ed.* 51, 7275–7278 (2012)). In these hydroxylation processes, the formed metal-active oxygen species, usually in the form of metal=O, metal-O-O· and metal bis(μ-oxo), were involved in an electrophilic attack on the benzene ring, resulting in high activity by using the mono-substituted benzenes with electron-donating groups such as anisole and phenol. By contrast, apparent high activity in the hydroxylation of benzene and suppression of the activity in the hydroxylation of anisole and phenol was observed using the VSZ-5 catalyst. Besides, for aryl halides substrates with electron-withdrawing groups, the high activity and chemo-selectivity were similarly obtained in the current study, which was rarely achieved in those previous electrophilic reactions. These phenomena suggest that our V-based hydroxylation process is different from those previous systems involving electrophilic attack (*Angew. Chem. Int. Ed.* 56, 12260–12263 (2017); *Angew. Chem. Int. Ed.* 56, 7779–7782 (2017); *J. Am. Chem. Soc.* 137, 5867–5870 (2015); *Chem. Commun.* 50, 11454–11457 (2014); *Angew. Chem., Int. Ed.* 51, 7275–7278 (2012)).

V-catalyzed oxidations usually involve the formation of peroxo vanadyl complexes (*Coordin. Chem. Rev.* 318, 135–157 (2016); *Coordin. Chem. Rev.* 257, 732–754 (2013); *J. Catal.* 267, 140–157 (2009); *Coordin. Chem. Rev.* 237, 89–101 (2003); *Chem. Rev.* 94, 625–638 (1994); *J. Am. Chem. Soc.* 105, 3101–3110 (1983)). In most cases, hydroperoxyl (HOO·) and hydroxyl (HO·) radicals were active sites (*Coordin. Chem. Rev.* 257, 732–754 (2013); *J. Catal.* 267, 140–157 (2009); *Coordin. Chem. Rev.* 237, 89–101 (2003); *Chem. Rev.* 94, 625–638 (1994)). Alternatively, oxygen transformation from V-peroxo complex to arenes was also achieved through the formation of metal-oxygen active sites, including both V-based electrophilic radicals and nucleophilic agents (*Coordin. Chem. Rev.* 318, 135–157 (2016); *J. Catal.* 267, 140–157 (2009); *Coordin. Chem. Rev.* 237, 89–101 (2003); *Angew. Chem. Int. Ed.* 42, 5063–5066 (2003); *Chem. Rev.* 94, 625–638 (1994); *J. Am. Chem. Soc.* 105, 3101–3110 (1983)).

In this work, structure characterization of the fresh VSZ-5 and the intermediate VSZ-5(m) indicated that the tetrahedral metavanadate V(V) species (VO₃)_nⁿ⁻ in the fresh catalyst changed to V(IV) species (**B**) after the addition of H₂SO₄ (details are provided in the seventh paragraph in the section “Catalytic activity: Hydroxylation of arenes”). Previous studies of the behavior of V species in the presence of H₂O₂ (*Coordin. Chem. Rev.* 318, 135–157 (2016); *Coordin. Chem. Rev.* 257, 732–754 (2013); *J. Catal.* 267, 140–157 (2009); *Coordin. Chem. Rev.* 237, 89–101 (2003); *J.*

Mol. Catal. A: Chem. 164, 181–189 (2000); *Chem. Rev.* 94, 625–638 (1994); *J. Am. Chem. Soc.* 105, 3101–3110 (1983)) suggested that the species (**B**) is readily converted into (**C**). The observed umbrella model V species in the recovered catalysts VSZ-5(r1) and VSZ-5(r6) as well as the characterization of VSZ-5(r1)-as support the formation of this V(IV)-based peroxo compound during the reaction. The structure characterization and activity assess of VSZ-5(r1) and VSZ-5(r1)-as implies that the arene was oxidized by the active metal-oxygen sites derived from species (**C**) in the presence of H₂O₂. The interaction of species (**C**) with H₂O₂ to create species (**D**) is preferred in a non-radical hydroxylation reaction (*Coordin. Chem. Rev.* 257, 732–754 (2013); *J. Catal.* 267, 140–157 (2009); details are provided in the Answer to Question 7 of Reviewer #2). The formed diperoxo group in species (**D**) is a highly active oxygen transfer agent to initiate the ring hydroxylation of arenes (*Coordin. Chem. Rev.* 318, 135–157 (2016); *Coordin. Chem. Rev.* 257, 732–754 (2013); *J. Catal.* 267, 140–157 (2009); *Chem. Rev.* 104, 849–902 (2004); *Coordin. Chem. Rev.* 237, 89–101 (2003); *J. Mol. Catal. A: Chem.* 164, 181–189 (2000); *Chem. Rev.* 94, 625–638 (1994); *J. Am. Chem. Soc.* 105, 3101–3110 (1983)).

Based on above, we propose the most probable reaction mechanism in Fig. 1a in the revised manuscript. V(V) species (**A**) of the fresh VSZ-*n* in CH₃CN was reduced to V(IV) species (**B**) in VSZ-5(m) upon addition of H₂SO₄. After introducing H₂O₂, (**B**) was converted into V(IV) peroxo complex (**C**), quickly *via* the two transition states marked in the prompting frame (*Coordin. Chem. Rev.* 318, 135–157 (2016); *Coordin. Chem. Rev.* 257, 732–754 (2013); *J. Catal.* 267, 140–157 (2009); *Chem. Rev.* 104, 849–902 (2004); *Coordin. Chem. Rev.* 237, 89–101 (2003); *J. Am. Chem. Soc.* 105, 3101–3110 (1983)). Further interaction of H₂O₂ with (**C**) created V(IV) diperoxo species (**D**) (*Coordin. Chem. Rev.* 257, 732–754 (2013); *J. Catal.* 267, 140–157 (2009); *J. Mol. Catal. A: Chem.* 164, 181–189 (2000)). The interaction of species (**D**) with arene forms a transition complex, the cleavage of which affords phenol and regenerates species (**C**).

According to this mechanism, the kinetic equation is given as follows:

$$d[\text{phenol}]/dt = k'[\text{Cat.}][\text{S}][\text{H}_2\text{O}_2]$$

[phenol], [Cat.], [S], and [H₂O₂] are the concentration of phenol, catalyst, substrate and H₂O₂ respectively; *t* is the reaction time; *k'* is the catalytic rate constant.

The influence of the concentration of substrate was investigated in the VSZ-5 catalyzed hydroxylation of benzene with H₂O₂ (Fig. R2). This result and previous investigation of the reaction conditions suggests that the reaction rate is proportional to concentration of VSZ-5, benzene and H₂O₂ (Fig. R3). The experimental kinetic equation is in line with that from the proposed mechanism, thus further supporting the above proposed mechanism.

In the present form, we tried our best to propose a most probable mechanism based on the characterization of the special active sites as well as the catalytic performance in the hydroxylation of arenes. In order to avoid potential confusion, we have rewritten the description of the proposed mechanism (Section “Discussion”) and improved the presentation of Fig. 1 in the revised manuscript. Corresponding KIE (Paragraph 1, Section “Discussion”) and kinetic analyses (Paragraph 3, Section “Catalytic activity: Hydroxylation of arenes”) have been added into the revised manuscript and Supplementary Methods (Figs R2 and R3 are added as Supplementary Figs 23 and 30-32). Nonetheless, we also realize that the details of the mechanism are still to be clarified. An unambiguous understanding of all mechanistic details of the new hydroxylation reaction requires a lot further additional efforts.

Figure R2 | Yield and selectivity as a function of amount of benzene in VSZ-5 catalyzed benzene hydroxylation with H_2O_2 . Reaction condition: benzene (5 mmol), aqueous H_2O_2 (30%, 5 mmol), VSZ-5, H_2SO_4 (0.15 g), CH_3CN (14 mL), 80 °C, 30 s. Yield (%): $[\text{phenol} (\text{mol})/\text{initial benzene} (\text{mol})] \times 100$. Selectivity (%): $[\text{phenol} (\text{mol})/\text{converted benzene} (\text{mol})] \times 100$.

Figure R3 | Linear plot of the reaction rate ($\text{d[PhOH]}/\text{dt}$) vs. concentration of (a) catalyst VSZ-5 (0.025–0.1 g), (b) H_2O_2 (1–3.3 mol) and (c) benzene (1–5 mol). Reaction conditions: benzene (5 mmol), aqueous H_2O_2 (30%), VSZ-5 (0.1 g), H_2SO_4 (0.15 g), CH_3CN (14 mL), 80 °C, 30 s. For each figure, there is a specific parameter changed. $\text{d[PhOH]}/\text{dt}$ was calculated from $[\text{PhOH}]/t$ at 30 s by assuming that the kinetic curve belongs to a first-order equation. The data in Fig. R2a and R2b were calculated from Supplementary Figs 21 and 22, respectively. The data in Fig. R2c were calculated from Fig. R2.

2. Authors used H_2O_2 with very less amount (arene/ H_2O_2 ration) in their reaction and comparatively conversion levels are very high in a very short time, So efficiency of H_2O_2 for this catalyst is quite high , which needs to be crosschecked.

Answer: Thanks for the important comment. In a typical run, VSZ-*n* catalyzed hydroxylation of arenes was carried out under the stoichiometric condition (arene/ $\text{H}_2\text{O}_2=1/1$, molar ratio), in which 5 mmol arene and 5 mmol H_2O_2 were used. H_2O_2 amount in our reactions is less than many hydroxylation systems by using excessive H_2O_2 , in which several to more than ten folds of H_2O_2 relative to arene was applied to afford sufficient conversion of the arene substrate. Herein, VSZ-*n* catalyzed aromatic ring hydroxylation resulted in comparable yield of the phenols by using equal amount of H_2O_2 to arene. Supplementary Tables 6 and 7 list the reaction conditions of the previous typical studies for the hydroxylation of benzene and toluene. The arene/ H_2O_2 ratios are 0.1-50 for those homogeneous catalysts and 0.056-50 for those heterogeneous ones. Compared with those previous systems, the amount of H_2O_2 used in this work (arene/ $\text{H}_2\text{O}_2=1/1$) is less than those involving high H_2O_2 /arene ratios (up to 18). On the other hand, some previous studies (Supplementary Tables 6 and 7) employed much less amount of H_2O_2 relative to substrate with very low H_2O_2 /arene ratios (down to 0.02), for perusing higher efficiency of H_2O_2 . However, this condition resulted in the lower yield toward the target product. Our stoichiometric condition (arene/ $\text{H}_2\text{O}_2=1/1$) is more practically desirable and useful by providing high atom-efficiency of both the substrate and oxidant.

The efficiency of H_2O_2 was calculated according to the following equation: efficiency of H_2O_2 = [products (mol)]/[initial H_2O_2 (mol)]. The efficiency of H_2O_2 for this work, as well as previous literatures, was presented as “ Y_o ” in Supplementary Tables 6 and 7. Calculation of the efficiency of H_2O_2 therefore relies on the quantitative measurement of the reaction product. The molar amount of aromatic hydroxylation product of this work was measured through GC and GC/MSD. As mentioned in the section “Catalytic activity evaluation”, the identification of products was analyzed by GC/MSD (Agilent Technologies 7890B-5977A GC/MSD) equipped with a capillary column (HP-5MS 30m \times 0.32mm \times 0.25 μ m) and a Triple-Axis detector. Quantitative analyses were carried out by gas chromatography (GC, Agilent 7890B) equipped with a flame ionization detector and a capillary column (HP-5, 30m \times 0.25mm \times 0.25 μ m). GC and GC/MSD spectra of a typical hydroxylation process are presented in Fig. R4. We are sorry for the lack of the above mentioned information in the original submission. According to this comment, Fig. R4 is added as Supplementary Fig. 67 in the revised Supplementary Methods. The corresponding description “ $Y_o(\%)=[\text{products (mol) /initial } \text{H}_2\text{O}_2 \text{ (mol)}]\times 100$ ” in the footnote of the Supplementary Tables 6 and 7 is changed to be “ $Y_o(\%)=[\text{products (mol) /initial } \text{H}_2\text{O}_2 \text{ (mol)}]\times 100$, also denoting the efficiency of H_2O_2 .” Indeed, the efficiency of H_2O_2 (Y_o) was 30.8% for VSZ-5 catalyzed benzene hydroxylation under stoichiometric condition (arene/ $\text{H}_2\text{O}_2=1/1$, molar ratio) (Supplementary Table 6). When the hydroxylation was carried out at H_2O_2 /arene=0.1 (Supplementary Fig. 22), even higher Y_o value of 43.5% was obtained, in consistent with the observation that a low amount of H_2O_2 used relative to arene usually enhances H_2O_2 efficiency. These results suggest high efficiency of H_2O_2 for this new hydroxylation process.

Figure R4 | (a) GC and (b-d) GC/MSD spectra of a typical hydroxylation of benzene catalyzed by VSZ-5.

3. Above all the colour change of the catalyst from white to green after addition of Conc. Sulphuric acid to the reaction mixture showing the leaching tendency of Vanadium species during the reaction. This indicates that the catalyst is not truly heterogeneous. The colour of the catalyst should be white for truly heterogeneous in nature.

Answer: Thanks for the comment. In order to confirm the heterogeneous nature of VSZ-5 catalyzed hydroxylation reaction, we investigated the activity of the filtrate in a hot-filtration test. Because the reaction is finished within such a short time, it is impossible to hot-filtrate the catalyst during the reaction. Our operation was slightly different from the traditional test. Details of the filtration test are as follows. The catalyst VSZ-5 (0.1 g) was mixed with benzene (5 mmol), CH₃CN (14 mL) and H₂SO₄ (0.15 g). After the slurry mixture was stirred at 80 °C for 30 minutes, the catalyst was removed by filtration. The reaction of the filtrate solution alone was carried out with the addition of 5 mmol H₂O₂ (30% aqueous solution) at 80 °C. No phenol was detectable and the filtrate was colorless. The result verifies that the hydroxylation process is intrinsically heterogeneous and excludes the contribution of the potentially leached V species.

The color change of the catalyst is attributable to the valent state variation of the V species. Structure characterization (Supplementary Figs 5 and 40-57) of collected intermediate VSZ-5(m) indicated that the V(V) species in the fresh VSZ-5 changed to V(IV) species in VSZ-5(m) after the addition of H₂SO₄. Owing to the change of valent state, the color of the catalyst changed to dark green in the reaction medium. Such variation has been observed for many previous V based heterogeneous catalysts (*J. Am. Chem. Soc.* 123, 12101-12102 (2001); *Catal. Sci. Technol.* 3, 1394–1404 (2013); *Chem. Eng. J.* 173, 620–626 (2011)). The color change of a solid catalyst has also been found in other metal-based catalysts such as WO₃, MoO₃ and TiO₂ (*Chem. Rev.* 95, 537–550 (1995); *Chem. Rev.* 98, 307–325 (1998); *J. Am. Chem. Soc.* 126, 10657-10666 (2004); *Angew. Chem.* 124, 3420–3423 (2012); *Angew. Chem.* 127, 440–445 (2015)).

In order to avoid potential confusion, the following description is added into the revised manuscript (Paragraph 1, Section “Catalytic activity: Hydroxylation of arenes”): “A hot-filtration test was performed by filtrating the solid catalyst after the addition of H₂SO₄. The filtrate solution is inactive in promoting benzene to phenol, verifying that the hydroxylation process is intrinsically heterogeneous and excludes the contribution of the potentially leached V species.” The experimental details are added into the section “Catalytic activity evaluation” as follows: “A hot-filtration test was performed to confirm the heterogeneous nature of the hydroxylation reaction. Because the reaction is finished within a short time, it is impossible to hot-filtrate the catalyst during the reaction. The operation was slightly different from the traditional test. Details are as follows. The catalyst VSZ-5 (0.1 g) was mixed with benzene (5 mmol), CH₃CN (14 mL) and H₂SO₄ (0.15 g). After the slurry mixture was stirred at 80 °C for 30 minutes, the catalyst was removed by filtration. The reaction of the filtrate solution alone was carried out with the addition of 5 mmol H₂O₂ (30% aqueous solution) at 80 °C. The products were measured by GC.”

Reviewer #2:

1. The reduced VO_3 (IV) seems to be considered as the real active sites. Then, a positive correlation between the number of the active sites and the reactivity. Perhaps by ESR?

Answer: According to the suggestion of the reviewer, we newly collected two intermediate samples of VSZ-1(m) and VSZ-3(m) (corresponding fresh catalysts are VSZ-1 and VSZ-3 respectively, in which 1 and 3 stand for $100 \times [\text{V}/\text{Si}]$ molar ratio in the gel), using the procedure similar to the one used to collect VSZ-5(m). ESR spectra of VSZ-1(m) and VSZ-3(m) were measured and compared with that of VSZ-5(m). As shown in Fig. R5, all the three samples exhibited apparent ESR signals derived from the formation of V(IV) species. Their peak intensities are in the sequence of $\text{VSZ-1(m)} < \text{VSZ-3(m)} < \text{VSZ-5(m)}$. The ESR observation suggests that more and more V(IV) species were formed as the total V content increased in the catalysts VSZ-1, VSZ-3 and VSZ-5, in line with their activity sequence for the hydroxylation of benzene (Supplementary Table 4, entry 4 (9.6%), 6 (17.6%) and 7 (30.8%)) and toluene (Supplementary Table 5, entry 3 (9.8%), 4 (13.4%) and 5 (26.2%)). This comparison reflects a positive correlation between the number of the V(IV) species and the reactivity. The following description is added into the revised manuscript (Paragraph 7, Section “Catalytic activity: Hydroxylation of arenes”): “This is further reflected by the positive correlation between the number of the V(IV) species and the reactivity (Supplementary Fig. 46).” Fig. R5 is added as Supplementary Fig. 46 in the revised Supplementary Methods. The details are added into the caption of this figure.

Figure R5 | ESR spectra of VSZ-1(m), VSZ-3(m) and VSZ-5(m).

2. Since the reaction is finished very quickly (30 S), then why not increase the amount of substrates to get a more precise TOF?

Answer: Thanks for the valuable suggestion. Indeed, we had investigated various reaction conditions, including the substrate concentration, H_2O_2 amount, catalyst amount, solvent type and amount, additive type and amount, reaction temperature and time (Supplementary Figs 18-28; Supplementary Tables 4 and 5). One of the purposes of this detailed investigation was to get a reasonably precise TOF. However, whenever the wanted high yield and selectivity were obtained, the reaction was finished within a short time (i.e., almost immediately within 30 s). Reducing the

reaction temperature or the catalyst dosage is usually an effective way to decrease the reaction rate but caused remarkably lower activities in our reaction system (Supplementary Figs 18, 21, 24, 27). As suggested by the reviewer, directly increasing the amount of substrate was also tried in VSZ-5 catalyzed hydroxylation of toluene (conditions: toluene (5~25 mmol), aqueous H₂O₂ (30%, 5~25 mmol), VSZ-5 (0.1 g), H₂SO₄ (0.15 g), CH₃CN (12 mL)). If without changing the H₂O₂ dosage (e.g. 25 mmol toluene: 5 mmol H₂O₂), then more or less the same TOF as that under the optimal condition (5 mmol toluene: 5 mmol H₂O₂) was obtained (Table R1, entry 1 and 2). If additional H₂O₂ was added (e.g. 25 mmol toluene: 25 mmol H₂O₂), declined activity toward the target phenol product was observable (Table R1, entry 3). We propose the following reason for the declined activity. When an excessive amount of H₂O₂ was used, larger amount of water was introduced into the reaction system, changing the solvent composition. The change of solvent composition could have hampered the ring-hydroxylation of arenes and caused the over-oxidation of product. Consequently, it is hard to get a more precise TOF by increasing the amount of substrates for this special hydroxylation process.

Table R1. Direct hydroxylation of toluene with H₂O₂.[†]

Entry	Catalyst	Toluene amount (mmol)	H ₂ O ₂ amount (mmol)	S (%) [‡]	Y (%) [§]	TOF (h ⁻¹) [*]
1	VSZ-5	5	5	91.8	26.2	1969
2	VSZ-5	25	5	89.0	5.4	2029
3	VSZ-5	25	25	32.4	1.5	563

[†]Reaction conditions: toluene (5 mmol), H₂O₂ (30%, 5 mmol), VSZ-5 (0.1 g), H₂SO₄ (0.15 g), CH₃CN (12 mL), 80 °C, 30 s. [‡]Selectivity for cresols: [cresols (mol)/converted toluene (mol)]×100. [§]Yield of cresols: [cresols (mol)/initial toluene (mol)]×100; equal molar ratio of *o*- and *p*-cresol forms while no *m*-cresol is detected. ^{*}Turnover frequency (TOF)=[cresols (mol)/(V (mol)×reaction time (h))].

3. The reaction time contributed by diffusion effect is notable considering the microporous structure of ZSM. Then the actual activity should well be higher than the offered value. With such a high reaction rate, it is hard to understand the non-radical mechanism. A apparent energy barrier is necessary as a reference. If the calculated energy barrier is lower than 30 KJ/mol. Then it means the diffusion is controlled. With such a high activity to C-H activation, it is hard to understand the extremely low activity to phenol over-oxidation in that the electron-withdrawing strength of OH group is between OCH₃ and H, as mentioned by the authors about the substituent effect in Table 1. **Answer:** According to the suggestion of the reviewer, the activation energy barrier in the VSZ-5 catalyzed hydroxylation of benzene/toluene is calculated according the Arrhenius Equation:

$$k=A*\exp(-Ea/RT),$$

(k: reaction rate; A: pre-exponential factor; Ea: activation energy; R: gas constant; T: reaction temperature)

Because the reaction was finished within a short time (30 s), we assume that the reaction belongs to a linear kinetic curve (d[phenol]/dt=k), just like the very initial stage of a conventional kinetic curve for the benzene hydroxylation with the reaction time in hours. Therefore, the reaction rate at different temperature is calculated from [phenol]/t, in which [phenol] is the phenol concentration after reaction while t is fixed as 30 s. According to this assumption and based on the

activity data in Supplementary Figs 18 and 24, the linear plot of $\ln k$ vs. $1/T$ was performed and the slope is $-E_a/R$ (Fig. R6). The calculated E_a is 24 and 26 kJ mol⁻¹ for the VSZ-5 catalyzed hydroxylation of benzene and toluene, respectively.

Figure R6 | Linear plot of $\ln k$ vs. $1/T$ in VSZ-5 catalyzed hydroxylation of (a) benzene and (b) toluene.

The above calculation reveals relatively low activation energies for the VSZ-5 catalyzed hydroxylation of benzene and toluene, in accordance with the rapid reaction rate in these hydroxylation processes. However, low E_a values (<30 kJ mol⁻¹) do not necessarily mean that the reaction was controlled by diffusion in our reaction systems. In the catalysis test, fine powders of catalyst below 200 meshes were used with the stirring rate higher than 1000 rpm, minimizing the possible diffusion limitation. In the exploration of substrate scope, similar activities were observed in converting different benzene derivatives with one or two substituted groups into the corresponding phenols. This is typically a reaction-controlled result. If it is a diffusion-controlled reaction, *o*- and *m*-xylene, larger in molecular size than *p*-xylene, should be inhibited in the conversion; nonetheless, *o*-, *m*- and *p*-xylenes afforded similar yields of 20.6-22.1%. Noticeably, the product distribution in the oxidation of bulkier substrates such as tert-butylbenzene and naphthalene was still affected by the microporous channels of zeolite, and *p*-product was preferentially formed. Further, in the investigation of reaction conditions for the VSZ-5 catalyzed hydroxylation of benzene and toluene (Supplementary Figs 18-28; Supplementary Tables 4 and 5), the dramatic change in activity along with these reaction conditions reveals that the hydroxylation is mostly controlled by the intrinsic properties of the active sites.

Though the electron-withdrawing strength of OH and OCH₃ groups is larger than H, these two groups are usually regarded as the electron-donating groups, attributable to that the resonance effect (electron-donating) of them is stronger than inductive effects (electron-withdrawing). According to the proposed mechanism (also please see details in Answers to Question 1 of Reviewer #1 and Question of Reviewer #3), the interaction between V(IV) species (**D**) and arene involves a nucleophilic attack of the metal-oxygen sites on aromatics to produce phenols. In such a case, the oxidation of benzene derivatives with strong electron-donating groups is suppressed, resulting in low activity for these substrates.

Fig. R6 is added as Supplementary Fig. 29 into the revised Supplementary Methods. The following description is also added into the revised manuscript (Paragraph 3, Section “Catalytic activity: Hydroxylation of arenes”): “The activation energy barrier in the VSZ-5 catalyzed hydroxylation of benzene/toluene is calculated according the Arrhenius Equation:

$k=A*\exp(-E_a/RT)$ (k : reaction rate; A : pre-exponential factor; E_a : activation energy; R : gas constant; T : reaction temperature). Because the reaction was finished within a short time (30 s), we assume that the reaction belongs to a linear kinetic curve ($d[\text{phenol}]/dt=k$), just like the very initial stage of a conventional kinetic curve for the benzene hydroxylation with the reaction time in hours. Therefore, the reaction rate at different temperature is calculated from $[\text{phenol}]/t$, in which $[\text{phenol}]$ is the phenol concentration after reaction while t is fixed as 30 s. According to this assumption and based on the activity data in Supplementary Figs 18 and 24, the linear plot of $\ln k$ vs. $1/T$ was performed and the slope is $-E_a/R$ (Supplementary Fig. 29). The calculated E_a is 24 and 26 kJ mol^{-1} for the VSZ-5 catalyzed hydroxylation of benzene and toluene, respectively. The relatively low activation energies are in accordance with the rapid reaction rate in these hydroxylation processes.”

4. As for Fig 3b and e, it seems unnecessary to prolong the reaction time since no additional oxidant (H_2O_2) is left and the reaction stop naturally.

Answer: We agree that H_2O_2 was barely left after several minutes’ reaction due to its contribution to hydroxylation and self-decomposition. Herein, prolonging the reaction time (up to the conventional reaction time in tens of minutes or hours) was applied to give an experimental proof that the hydroxylation of arenes preceded within a short time, while prolonging time would not induce a decline of phenol yield. This is distinctively different from previous “volcanic” type kinetic curves that implied over-oxidation (*Green Chem.* 18, 5643–5650 (2016); *Catal. Sci. Technol.* 3, 1394–1404 (2013)).

5. As for the radical scavenger tests, can they diffuse into the micro-pores?

Answer: Thanks for the valuable question of the reviewer. We performed the optimization of the molecular structure of all the radical scavengers we used (tert-butyl alcohol (TBA), butylated hydroxytoluene (BHT), benzoquinone (BQ), 5,5-Dimethyl-1-pyrroline N-oxide (DMPO), and bromotrichloromethane (BrCCl_3)). The sizes of the optimal structures for these radical scavengers are compared with that of the ten-member-ring of ZSM-22 (Fig. R7). The size of the ten-member-ring of all-silica ZSM-22 was obtained by referring to the data base of “Materials studio” and “Database of zeolite structure” (<http://www.iza-structure.org/databases/>). Pore size ($6.3 \text{ \AA} \times 7.9 \text{ \AA}$) of all-silica ZSM-22 is larger than that of Al-containing analogue (normally $4.5 \text{ \AA} \times 5.5 \text{ \AA}$), as the latter has counter cations that decrease the pore diameter. DFT (Density Functional Theory) calculation for the optimization of these radical scavengers was carried out on using Gaussian 09 programs at the B3LYP level of theory with 6-31G basis sets. Cartesian coordinates for the optimized geometries were listed in the Appendix at the end of this “Responses to referees”. The size of each atom was included in the measurement of the molecular size and diameter of ten-member-ring of all-silica ZSM-22.

The results show that the scavengers of BQ, TBA, BrCCl_3 and DMPO are small molecules that can diffuse into the micro-pores of all-silica ZSM-22. The width of BHT ($4.9 \text{ \AA} \times 7.1 \text{ \AA}$) is smaller than the diameter of ten-member-ring of ZSM-22, suggesting that even this bulky scavenger may enter the ZSM-22 pores.

Figure R7 | Size of (a) ten-member-ring of all-silica ZSM-22 and optimized structure of radical scavengers: (b) BHT, (c) DMPO, (d) TBA, (e) BrCCl₃, and (f) BQ.

6. For the step from (B) to (C) in Fig 1, why H₂O can serve as a stable ligand without being abstracted by one H by the atop O binding with V, namely the usual hydrogen transfer effect?

Answer: Thanks for the insightful comment. According to the reported crystal structure of peroxido complexes of vanadium (*Coordin. Chem. Rev.* 318, 135–157 (2016); *Coordin. Chem. Rev.* 237, 89–101 (2003); *Chem. Rev.* 104, 849–902 (2004); *Chem. Rev.* 94, 625–638 (1994)), H₂O can serve as a stable ligand in numerous peroxidovanadium species, in which there co-existed V=O and H₂O. No hydrogen transfer from water to atop O of V=O was observed in those complexes. Because the H-abstraction process needs to overcome the high bond dissociation energy (BDE) of O-H in water (>450 kJ mol⁻¹), such H-abstraction is energy unfavorable. Therefore, H₂O could reasonably serve as a stable ligand in the V species (B). Water is only weakly coordinated to the V species, and the solvent CH₃CN may also serve as the ligand of the V species (B) replacing water. That is why we adopted “L” to denote the ligand (water or CH₃CN) in the V species (B). In order to give a concise and clear layout, and to avoid potential confusion, the “L” is ignored in the revised Fig. 1. For the same purpose, the short bond linking V and O atoms is denoted as the commonly used V=O replacing initial V≡O in the revised Fig. 1.

7. For the step from (C) to (D) in Fig 1, the authors should give a more detailed description about the process since I cannot map this route without radical pathway.

Answer: Prompt by this suggestion, the details for the step from (C) to (D) are proposed as follows (Fig. R8). The variation from species (C) to (D) included the formation of a H₂O₂-V^{IV} complex through the interaction of species (C) with H₂O₂, followed by H-transfer to the oxo-ligand and elimination of H₂O.

It was reported that two possible paths may happen after the steps of the interaction of V^V -peroxo with H_2O_2 and the H-transfer (*Coordin. Chem. Rev.* 257, 732–754 (2013); *J. Catal.* 267, 140–157 (2009); *J. Mol. Catal. A: Chem.* 164, 181–189 (2000)): (1) Elimination of H_2O causing the formation of diperoxo-V species; (2) Elimination of $HOO\cdot$ yielding V^{IV} species. Previous work indicated that these two paths had similar activation energy barriers, but formed diperoxo-V species were more stable than V^{IV} species (*Coordin. Chem. Rev.* 257, 732–754 (2013); *J. Catal.* 267, 140–157 (2009)). In this work, mechanisms involving free radicals were excluded based on the catalytic performance in the hydroxylation of different arenes (benzene and its derivatives with electron-donating and withdrawing groups), KIE analysis, and radical scavenger tests (details are seen in the third paragraph of the Answer to Question 1 of Reviewer #1). Therefore, we propose that the elimination of H_2O is the most probable way in the variation from species (C) to (D) (Fig.R8).

The following description (Paragraph 3, Section “Discussion”) is added into the revised manuscript: “Further interaction of H_2O_2 with (C) created $V(IV)$ diperoxo species (D) through the formation of a H_2O_2 - V^{IV} complex, followed by H-transfer to the oxo-ligand and elimination of H_2O (Supplementary Fig. 65)^{44,57,59}.” Fig. R8 is added as Supplementary Fig. 65 in the revised Supplementary Methods. The details are added into the caption of this figure.

Figure R8 | Formation of species (D) through the interaction of species (C) with H_2O_2 .

Reviewer #3:

This is an interesting report. The system reported by the authors seem to be highly effective and selective for the conversion of benzene to phenol, an important industrial reaction. While I am not an expert on the characterization of the catalyst my analysis of the data would suggest that this system is unusually active compared to other reports. While the author makes such a comparison with respect to TOF, selectivity, yields, etc., I was perhaps most impressed by the volumetric productivity, Space Time Yield (STY). Assuming 5 mmol of product in 14 ml over 30 s I calculate a STY of $\sim 10E^{-5}$ mol/cc.s. This is 10 times higher than the typical STY of many commercial processes based on the so-called Weiss window of 10^{-6} mol/cc.s. Initially, I was concerned that H₂SO₄ additive was responsible for the catalysis but the authors address this. The only issue I have with this report is that the suggested basis for the selectivity does not seem reasonable. It is remarkable that phenol and anisol are less reactive than benzene. This is certainly good evidence against a radical mechanism. However, other metal-based systems, including those based on V, do not show such remarkable selectivity. The authors suggest that this is due to the O on the active V(IV)O species being “negative” and that OMe and OH are donor groups that inhibit attack on phenol or anisol. This does not seem reasonable. Nucleophilic attacks on aromatics leading to oxidation typically require nitro groups to activate the arene. It is unlikely that benzene would enter in such a nucleophilic reaction. In any case, while the O in V(IV)-O is formally 2-, it is not considered a nucleophile and likely reacts as an O-electrophilic. Such species have been proposed for Os, Cu and V based on homogeneous systems. Additionally, while OMe and OH are pi-donors (they facilitate aromatic electrophilic substitution through resonance not through inductive effects) they are sigma-acceptors (more electron-withdrawing than H) and should activate anisol or phenol to nucleophilic reaction. This unusual lack of reaction with phenol bothers me but I cannot see an issue based on the data. I was struck by what seems to be a considerable amount of “hype” in the report. I recommend this be removed as it is distracting at best and over-selling at worst. I will leave that up to the editor. I recommend publication.

Answer: Thanks for the positive comments of the reviewer and the recommendation of publication. In this work, we report the VSZ-*n* catalyzed hydroxylation of arenes with H₂O₂. High yields were achieved under the stoichiometric condition (arene/H₂O₂=1/1) within a short time. Various benzene derivatives were oxidized into the corresponding hydroxylation products. The radical mechanism is excluded based on the catalytic performance of VSZ-5 in the hydroxylation of arenes (relative reactivity and region-selectivity in the hydroxylation of benzene and its substituted derivatives), KIE analysis and radical scavenger tests. As an alternative, a probable non-radical mechanism is proposed.

As pointed out by the reviewer, electrophilic metal-oxygen species, such as metal=O, metal-O-O· and metal bis(μ-oxo), have been proposed in several non-radical hydroxylation reactions (*Angew. Chem. Int. Ed.* 56, 12260–12263 (2017); *Angew. Chem. Int. Ed.* 56, 7779–7782 (2017); *J. Am. Chem. Soc.* 137, 5867–5870 (2015); *Chem. Commun.* 50, 11454–11457 (2014); *Angew. Chem., Int. Ed.* 51, 7275–7278 (2012)). In those previous systems, high activity was observed by using the mono-substituted benzenes with electron-donating groups such as anisole and phenol. By contrast, apparent high activity in the hydroxylation of benzene and suppression of the activity in the hydroxylation of anisole and phenol were observable in our system. Besides, for aryl halides substrates with electron-withdrawing groups, the high activity and chemo-selectivity were similarly obtained in the current study, which was rarely reached in those previous

electrophilic reactions. These phenomena suggest that our V-based hydroxylation process is different from those previous systems involving electrophilic attack (*Angew. Chem. Int. Ed.* 56, 12260–12263 (2017); *Angew. Chem. Int. Ed.* 56, 7779–7782 (2017); *J. Am. Chem. Soc.* 137, 5867–5870 (2015); *Chem. Commun.* 50, 11454–11457 (2014); *Angew. Chem., Int. Ed.* 51, 7275–7278 (2012)).

As discussed in the Answer to Question 1 of Reviewer #1 and Answer to Question 7 of Reviewer #2, in this work, the tetrahedral metavanadate V(V) species (VO_3)_nⁿ⁻ in the fresh catalyst changed to V(IV) species (**B**) after the addition of H₂SO₄, and then species (C) in the presence of H₂O₂. Further interaction of species (C) with H₂O₂ causes species (D), which is the most probable metal-active oxygen site for a non-radical hydroxylation reaction. The reaction of V-peroxo compound with arene to produce phenol may undergo either homolysis or heterolysis of V-O bond (*J. Catal.* 267, 140–157 (2009); *Coordin. Chem. Rev.* 237, 89–101 (2003); *Chem. Rev.* 94, 625–638 (1994); *J. Am. Chem. Soc.* 105, 3101–3110 (1983)). For conventional V^V-peroxo complex, heterolysis will generate V^{VI}-O-O⁻ and then V^V-O⁺-O⁻ species. Comparatively, homolysis of the V-O in V^V-peroxo to V^{IV}-O-O[·] is more likely in the hydroxylation of arenes (*Coordin. Chem. Rev.* 237, 89–101 (2003); *Chem. Rev.* 94, 625–638 (1994); *J. Am. Chem. Soc.* 105, 3101–3110 (1983)). Nonetheless, the behavior of the (low valent) V^{IV}-peroxo compound in this work is different from the conventional V^V-peroxo compound. Heterolysis of V^{IV}-peroxo species produces the V^V-O-O⁻ species, whereas homolysis may result in V^{III}-O-O[·]. Based on 1) the radical hydroxylation route is experimentally excluded; 2) the switch between V^{IV} and V^V is more energy favorable for the H₂O₂-mediated hydroxylation (*Coordin. Chem. Rev.* 257, 732–754 (2013); *J. Catal.* 267, 140–157 (2009); *Coordin. Chem. Rev.* 237, 89–101 (2003); *Chem. Rev.* 94, 625–638 (1994); *J. Am. Chem. Soc.* 105, 3101–3110 (1983)); 3) metal-oxygen species have also been reported to be involved in some nucleophilic attack processes (despite that they are more usually considered as electrophilic agents) (*Coordin. Chem. Rev.* 318, 135–157 (2016); *J. Catal.* 267, 140–157 (2009); *Coordin. Chem. Rev.* 237, 89–101 (2003); *Angew. Chem. Int. Ed.* 42, 5063–5066 (2003); *Chem. Rev.* 94, 625–638 (1994); *J. Am. Chem. Soc.* 105, 3101–3110 (1983)), we tentatively propose that a heterolytic cleavage of V-O in species (**D**) may occur in the oxidation of benzene ring (Fig. R9). This special behavior of the formation of V(IV) peroxo compounds leads to a different oxidation process from traditional V(V) peroxo. Thus formed V^V-O-O⁻ species are different from the previous metal=O and metal-O-O[·] species (*Angew. Chem. Int. Ed.* 56, 12260–12263 (2017); *Angew. Chem. Int. Ed.* 56, 7779–7782 (2017); *J. Am. Chem. Soc.* 137, 5867–5870 (2015); *Chem. Commun.* 50, 11454–11457 (2014); *Angew. Chem., Int. Ed.* 51, 7275–7278 (2012)). The V^V cation of V^V-O-O⁻ is a Lewis acidic site and able to interact with the negative π -system of the benzene ring *via* polarization. Such polarization interaction promotes the approaching and adsorption of arene, and the successive nucleophilic attack of O⁻ species on the C and H to form a transition complex, the cleavage of which produces phenol and regenerates species (C). In a traditional electrophilic substitution reaction of arenes, the following two steps are crucial: 1) an electrophilic agent attacks the C atom to form carbocation or a transition complex; 2) a nucleophilic agent attacks the formed carbocation or transition complex to afford the product. Similarly, the interaction of V^V-O-O⁻ with benzene ring involves the electrophilic attack of V^V cation on the benzene ring and the nucleophilic attack of O⁻ species on the C and H (this oxidation step is an oxygen transfer process, in which the O simultaneously interacts with C and H, enabling the H transfer from C to O). Owing to that, the reaction still involves the electrophilic attack of V^V

cation on the C in benzene ring, and *o*- and *p*-products were obtained. In the presence of an electro-donating group, the nucleophilic attack of O⁻ species on the electron-enriched C atom may be hindered, which decreases the activity in this step. Methoxy and hydroxyl group are electron-donating groups as they have the strong resonance effect (though with the coexistence of weak inductive effect). Thus, the hydroxylation of anisole and phenol is not favored through the proposed mechanism above, in line with the experimental results.

In this work, we tried our best to give as much insights as possible into the reaction mechanism. However, we realize that the proposed mechanism is only a probable one that best explain most reaction and spectroscopic observations. The inertness of hydroxylation of phenol is still unclear, as the electron-donating property of hydroxyl group is normally recognized as close to that of methoxy group. Additional studies are required in the future to clarify the unexpected behavior. Following the suggestion of the reviewer and in order to avoid potential confusion, we rewrite the description of this mechanism in the revised manuscript (Section “Discussion”). Fig. R9 and the above corresponding description are added as Supplementary Fig. 66 in the revised Supplementary Methods.

As a response to reviewer’s comment that the earlier version contains a considerable amount of “hype”, we removed or modified related statements to avoid distracting readers.

Figure R9 | Heterolysis of species (D) and successive interaction with arene to produce phenol.

Thank you for your kind considerations.

Yours sincerely,
Dr. Jun Wang

Appendix

Cartesian coordinates for the optimized geometries

Ten-member-ring of ZSM-22

O	3.75856000	6.60044000	3.34473000
Si	11.01930000	9.54964000	1.14111000
Si	9.77752000	12.38910000	0.72799100
O	8.23502000	12.58420000	1.04186000
O	10.10040000	10.81960000	0.82573000
O	10.68810000	9.05840000	2.63689000
Si	2.83971000	7.87036000	3.66011000
Si	4.08148000	5.03090000	3.24699000
O	5.62398000	4.83579000	3.56086000
O	3.75856000	10.81960000	0.82573000
O	3.17094000	9.05840000	2.63689000
Si	11.01930000	7.87036000	3.66011000
Si	9.77752000	5.03090000	3.24699000
O	8.23502000	4.83579000	3.56086000
O	10.10040000	6.60044000	3.34473000
Si	2.83971000	9.54964000	1.14111000
Si	4.08148000	12.38910000	0.72799100
O	5.62398000	12.58420000	1.04186000
Si	6.92950000	13.47780000	1.25950000
Si	6.92950000	3.94215000	3.77850000

BQ

C	0.00000000	1.26582700	-0.67414600
C	0.00000000	1.26582700	0.67414600
C	0.00000000	0.00000000	1.44125900
C	0.00000000	-1.26582700	0.67414600
C	0.00000000	-1.26582700	-0.67414600
C	0.00000000	0.00000000	-1.44125900
H	0.00000000	2.17955400	-1.25881500
H	0.00000000	2.17955400	1.25881500
H	0.00000000	-2.17955400	1.25881500
H	0.00000000	-2.17955400	-1.25881500
O	0.00000000	0.00000000	-2.69316900
O	0.00000000	0.00000000	2.69316900

TBA

C	0.46451100	-0.72991900	1.27002500
C	0.00597000	0.00535100	0.00000000
H	0.08128500	-0.21947500	2.15911600
H	0.09308000	-1.76317500	1.27728800
H	1.55829800	-0.76877100	1.33241600
C	0.46451100	-0.72991900	-1.27002500
C	0.46451100	1.46583500	0.00000000
H	0.08128500	-0.21947500	-2.15911600
H	1.55829800	-0.76877100	-1.33241600
H	0.09308000	-1.76317500	-1.27728800
H	1.55822600	1.52921900	0.00000000
H	0.08018500	1.97983500	-0.88630600
H	0.08018500	1.97983500	0.88630600
O	-1.46691900	0.09439200	0.00000000
H	-1.84558800	-0.80927200	0.00000000

BrCCl₃

C	0.00000000	0.00000000	-0.41363900
Cl	0.00000000	1.74604600	-1.03060100
Cl	-1.51212000	-0.87302300	-1.03060100
Cl	1.51212000	-0.87302300	-1.03060100
Br	0.00000000	0.00000000	1.57264200

DMPO

C	1.82040700	-0.78914200	0.09059100
C	1.63708000	0.72535500	-0.11238900
C	-0.61217600	-0.38126900	-0.02832900
C	0.48694500	-1.40477500	-0.41174300
H	2.68499000	-1.18062100	-0.45269600
H	1.96779600	-1.01290300	1.15331900
H	1.99405800	1.06335700	-1.09508300
H	2.11799200	1.34034300	0.65400800
H	0.29446000	-2.39073400	0.02183800
H	0.51683800	-1.51949200	-1.50243300
C	-1.74862000	-0.27521700	-1.05519100
H	-2.35105600	-1.19041600	-1.05870500
H	-2.39319600	0.57278200	-0.80542300
H	-1.34854600	-0.11636400	-2.06229500
C	-1.17701300	-0.61334500	1.38903500
H	-1.77320800	0.25374800	1.69032700
H	-1.81451400	-1.50451700	1.40750400
H	-0.37260700	-0.74715500	2.12094800

N	0.17178500	0.90002900	-0.02866200
O	-0.39565600	2.07026500	0.11218500

BHT

C	-0.00721600	2.29061500	-0.00724100
C	-1.20951100	1.57105000	-0.00649500
C	-1.25171800	0.17021500	0.00005000
C	-0.00851800	-0.50884000	0.00464000
C	1.23382100	0.16989400	-0.00017000
C	1.19275300	1.57550800	-0.00712800
H	-2.13685300	2.12966700	-0.01235400
H	2.11822800	2.13676100	-0.01342600
C	-0.01320100	3.80459500	0.00961600
H	-0.74747000	4.21203600	-0.69559200
H	-0.26945000	4.19658200	1.00357700
H	0.96813200	4.21034100	-0.25838800
C	-2.60220500	-0.58928400	-0.00095900
C	-3.80785000	0.38378600	-0.00731300
C	-2.72055800	-1.47769300	-1.27165400
C	-2.72713900	-1.46896500	1.27516600
H	-3.82087000	1.02708200	0.88019100
H	-3.81686700	1.02041700	-0.89966500
H	-4.73507000	-0.20015600	-0.00727400
H	-1.92421700	-2.22213900	-1.30865000
H	-3.68739000	-1.99712100	-1.27085500
H	-2.66844800	-0.86095400	-2.17728800
H	-3.69403700	-1.98826500	1.27288700
H	-1.93109400	-2.21320400	1.32130800
H	-2.67962200	-0.84613100	2.17687000
C	2.59416400	-0.58036800	-0.00103400
C	2.75316500	-1.44169100	1.28900200
C	2.74561000	-1.45329300	-1.28409000
C	3.79476300	0.40058500	-0.00866600
H	2.73930100	-0.79403600	2.17230200
H	1.96357800	-2.18659300	1.43347100
H	3.71157900	-1.97401900	1.26855000
H	2.72633200	-0.81362100	-2.17308900
H	3.70422800	-1.98531900	-1.26467700
H	1.95501700	-2.19923600	-1.41651700
H	4.72763800	-0.17445300	-0.00858200
H	3.79465200	1.03574600	-0.90076000
H	3.79942800	1.04388400	0.87756300
O	-0.08593400	-1.91346700	0.01032100
H	0.79258100	-2.32621400	0.01549600

Reviewers' Comments:

Reviewer #1 (Remarks to the Author):

The authors addressed almost all the issues raised by the reviewers and I recommend this manuscript for publication.

Reviewer #2 (Remarks to the Author):

After adding one more datum based on temperature in Supplementary Figure 29 a, I agree it to be published.

Reviewer #3 (Remarks to the Author):

The response of the author to my concerns are acceptable. I'm still not comfortable with the proposed mechanism as the details for the formation of the C-O and O-H bonds in the phenol product are not fully described. However, the author is entitled to speculate and, importantly, has added comments to make it clear to the reader that this is a proposal and more work is required. Considering the results and these changes, I'm in favor of publication.